

# Functional and evolutionary perspectives on gill structures of an obligate air-breathing, aquatic snail

Cristian Rodriguez[1,2,3,*], Guido I. Prieto[3,*], Israel A. Vega[1,2,3] and Alfredo Castro-Vazquez[1,2,3]

[1] IHEM, CONICET, Universidad Nacional de Cuyo, Mendoza, Argentina
[2] Universidad Nacional de Cuyo, Facultad de Ciencias Médicas, Instituto de Fisiología, Mendoza, Argentina
[3] Universidad Nacional de Cuyo, Facultad de Ciencias Exactas y Naturales, Departamento de Biología, Mendoza, Argentina
* These authors contributed equally to this work.

Corresponding authors
Israel A. Vega,
israel.vega7@gmail.com
Alfredo Castro-Vazquez,
a.castrovazquez@gmail.com

## ABSTRACT

Ampullariids are freshwater gastropods bearing a gill and a lung, thus showing different degrees of amphibiousness. In particular, *Pomacea canaliculata* (Caenogastropoda, Ampullariidae) is an obligate air-breather that relies mainly or solely on the lung for dwelling in poorly oxygenated water, for avoiding predators, while burying in the mud during aestivation, and for oviposition above water level. In this paper, we studied the morphological peculiarities of the gill in this species. We found (1) the gill and lung vasculature and innervation are intimately related, allowing alternation between water and air respiration; (2) the gill epithelium has features typical of a transporting rather than a respiratory epithelium; and (3) the gill has resident granulocytes within intraepithelial spaces that may serve a role for immune defence. Thus, the role in oxygen uptake may be less significant than the roles in ionic/osmotic regulation and immunity. Also, our results provide a morphological background to understand the dependence on aerial respiration of *Pomacea canaliculata*. Finally, we consider these findings from a functional perspective in the light of the evolution of amphibiousness in the Ampullariidae, and discuss that master regulators may explain the phenotypic convergence of gill structures amongst this molluscan species and those in other phyla.

## INTRODUCTION

Respiratory organs are identified as either gills or lungs, whether they are formed as protrusions or as invaginations of the respiratory mucosae. Gills are almost always used for aquatic respiration while lungs are for aerial respiration, but in addition to respiration, these organs may also serve other functions (*Maina, 2000a*, *2002b*).

In bimodal breathers (i.e. aquatic animals that have retained a gill while developing a respiratory organ for breathing air), the gill may partially lose the respiratory role while

acquiring others. In such cases, the respiratory function is supplied mainly by the air-breathing organ. In bimodal crustaceans and fishes, for example, the dependence on water comes in grades, the lesser water-dependent species having well-developed lungs that take up oxygen from the air and reduced or modified gills for ionic/osmotic regulation and $CO_2$ excretion (*Farrelly & Greenaway, 1987*; *Graham, Lee & Wegner, 2007*; *Hughes & Morgan, 1973*; *Innes & Taylor, 1986*; *Low, Lane & Ip, 1988*).

Most gastropods are marine and bear ctenidial gills (*Haszprunar, 1988*), but some have adapted to terrestrial and amphibious life by developing lungs as specialisations of the pallial cavity (*Semper, 1881*). Lungs occur in the subclasses Neritimorpha, Caenogastropoda and Heterobranchia (sensu *Bouchet et al., 2017*), although differing substantially in structure (*Lindberg & Ponder, 2001*). In the majority of lung-bearing gastropods, the pallial cavity itself has been modified as a lung (*Lindberg & Ponder, 2001*). Therefore, in these cases, the pallial cavity and the lung are homologous structures (*Ruthensteiner, 1997*). In turn, in the family Ampullariidae (Caenogastropoda), the lung is a sac with a single cavity that extends into the roof of the pallial cavity and thus is not homologous with the pallial cavity.

Ampullariids comprise bimodal breathers that have also retained a true gill (i.e. a ctenidium). As other bimodal breathers, ampullariids show different degrees of amphibiousness, being the genera *Afropomus*, *Saulea*, *Lanistes*, *Asolene*, *Felipponea* and *Marisa* more water-dependent than *Pila* and *Pomacea* (*Hayes et al., 2009a*). Amongst the latter, *Pomacea canaliculata* (Lamarck, 1822) is an obligate air-breather (*Seuffert & Martín, 2010*) that has a well-developed lung and a left nuchal lobe that is capable of being rolled into a siphon-like tube, and which uses as a snorkel to ventilate the lung while being submerged (*Andrews, 1965*). Behavioural observations have shown that *Pomacea canaliculata* relies mainly or solely on the lung for dwelling in poorly oxygenated waters, for avoiding predators (*Ueshima & Yusa, 2015*), while burying in the mud during aestivation (*d'Orbigny, 1847*; *Giraud-Billoud et al., 2011*, *2013*; *Hayes et al., 2015*), and for oviposition above the water level (*Hayes, Cowie & Thiengo, 2009b*). These facts make *Pomacea canaliculata* (and the Ampullariidae in general) interesting models to investigate the suitability of respiratory organs, that is, the gill and the lung, for the exchange of gases as well as the structural and functional integration of both organs. In a far-reaching perspective, such studies can help us understand the evolution of amphibiousness.

However, there is a paucity of data regarding the fine structure of caenogastropod gills. Previous studies have been limited to the hypertrophied ctenidia of hot vent caenogastropods (Provannidae; *Endow & Ohta, 1989*; *Stein et al., 1988*; *Windoffer & Giere, 1997*), which bear large quantities of endosymbiotic bacteria, but these studies have been aimed at elucidating the relationship between gill epithelial cells and their endosymbionts. In the Ampullariidae, the gill structure has been studied in *Pila globosa* (Swainson, 1822), *Marisa cornuarietis* (Linnaeus, 1758) and *Pomacea canaliculata* (*Andrews, 1965*; *Lutfy & Demian, 1965*; *Prashad, 1925*), but only at the light microscopy level. In this paper, we present a thorough description of the gill of *Pomacea canaliculata* at the anatomical (3D rendering of its blood system) and ultrastructural levels.

Surprisingly, gill structures have never been investigated in this respect with fine structural techniques before. Therefore, our findings are a significant contribution to the knowledge of gill structures of ampullariids and the extremely large group of caenogastropods in general. Also, we discuss the significance of our findings from a functional and evolutionary perspective.

## MATERIALS AND METHODS

### Animals and culturing conditions

Animals (young adult males, 20 mm shell length) were obtained from the Rosedal strain of *Pomacea canaliculata*, whose origin and culture conditions have been described several times elsewhere (*Cueto et al., 2015*; *Rodriguez et al., 2018*). Animals were immersed in water at 4 °C for 20–30 min both for relaxation and minimising pain, before careful shell cracking. Procedures for snail culture, sacrifice and tissue sampling were approved by the Institutional Committee for the Care and Use of Laboratory Animals (Comité Institucional para el Cuidado y Uso de Animales de Laboratorio, Facultad de Ciencias Médicas, Universidad Nacional de Cuyo), Approval Protocol No 55/2015.

### Light and electron microscopy

The gills from six animals were dissected out, fixed in dilute Bouin's fluid (1:2), dehydrated in a graded ethanol series, cleared in xylene, and embedded in a 1:1 paraffin–resin mixture (Histoplast®, Biopack, Buenos Aires, Argentina). Sections (three to five μm thick) were obtained and stained with Gill's haematoxylin and eosin. The stained sections were examined and photographed under a Nikon Eclipse 80i microscope using Nikon DS-Fi1-U3 camera and Nikon NIS-ELEMENT Image Software for image acquisition.

Additionally, the gill, lung and pericardium were dissected out as a single piece from two animals and were used for 3D reconstruction of the gill's blood system and for descriptions of its innervation. For these purposes, the lung was collapsed before fixation in dilute Bouin's fluid to reduce the size of the dissected sample. Then, samples were dehydrated, cleared and embedded as described above. Serial sections (10 μm thick) were stained with Gill's haematoxylin and eosin and photographed with a Nikon Digital Sight DS-5M camera on a Nikon Alphaphot-2 YS2 microscope.

Also, gill samples from six animals, each one including ~20 consecutive leaflets, were prepared for scanning electron microscopy. For this purpose, the samples were fixed in dilute Bouin's fluid and some of them were microdissected to show different aspects of the leaflets. Afterwards, they were dehydrated in an ethanol series, passed through acetone and then critical point dried, mounted on aluminium stubs, coated with gold, and examined with a Jeol/EO JSM-6490LV scanning electron microscope.

Furthermore, gill samples from six additional animals were fixed in Karnovsky's fluid (4% paraformaldehyde, 2.5% glutaraldehyde, dissolved in 0.1M phosphate buffer, pH 7.4). One day later, tissues were washed thrice in phosphate buffer and transferred to 1% osmium tetroxide overnight. Afterwards, they were dehydrated in a graded acetone series and finally embedded in Spurr's resin. Ultramicrotome sections (~200 nm) were stained with toluidine blue and covered with DPX medium (Sigma-Aldrich, St. Louis,
MO, USA) for fine histology. Silver-grey, ultrathin sections of gill samples were mounted on copper grids, stained with uranyl acetate and lead citrate, and examined with a Zeiss EM 900 transmission electron microscope.

## Computerised 3D rendering of the gill blood system

Digital images of every fifth section were aligned manually using Reconstruct, version 1.1.0.0 (*Fiala, 2005*), downloaded from Synapse Web, Kristen M. Harris, PI (http://synapses.clm.utexas.edu). Then, the profile of identified structures (*objects*) were drawn with the mouse using the Trace tool. These *traces* were used for the 3D visualisation of object structures. The 3D model was exported as a VRML 2.0 file and embedded in a PDF as described by *Ruthensteiner & Heß (2008)*, using the 3D tool of Adobe Acrobat 9 Pro Extended software.

## RESULTS

### General organisation of the gill and related pallial organs

The respiratory organs in the pallial complex are the gill and the lung. In addition, there are other organs that serve an accessory function, namely the siphon (= left nuchal lobe), the right nuchal lobe, and the pallial fold, which are depicted in Fig. 1A.

The gill extends from the left rear end of the mantle cavity (in the proximity of the pericardium and next to the ureter) to the right front side of the mantle cavity (close to the anus and the copulatory apparatus), and borders the posterior and right sides of the lung (Fig. 1A). The gill is formed by a single row of rather parallel leaflets (i.e. a ctenidial monopectinate condition) that hang from their bases in the roof of the mantle cavity (Fig. 1B). The gill leaflets have a rather triangular shape with free edges of unequal length (Fig. S1). Relative to their orientation in the pallial cavity, the shorter free edge is referred here to as the afferent border, and the longer to as the efferent one. The basal border is anchored to the branchial base, where the main afferent and efferent vessels run, and therefore to the inner mantle.

The lung extends along the roof of the mantle cavity and communicates with the mantle cavity through the pneumostome, close to the base of the siphon. The pallial fold is a mucosal ridge that extends on the floor of the mantle cavity. It originates at the left posterior end of the mantle cavity, close to the pericardium, and runs to the right until it crosses the prostate (or the vagina in females) and then takes a diagonal anteroposterior direction towards the proximity of the right nuchal lobe. A functionally significant narrow channel is delimited between this fold, the gill, and the rear wall of the mantle cavity. The right nuchal lobe appears in fixed specimens as a short mucosal triangular fold hanging from the right side of the neck (Fig. 1A). In living animals, however, it is a thin scoop-like structure, which may occlude partly or totally the excretory mantle opening.

### Blood circulation in the gill

The 3D rendering of the blood system of the gill (Figs. 2A–2C; Fig. S2) shows the *afferent branchial vessel* collects blood from ureteral efferents, particularly the *efferent ureteral vessel* (= *efferent renal vein* in *Andrews, 1965*). Additionally, other gill afferents come from

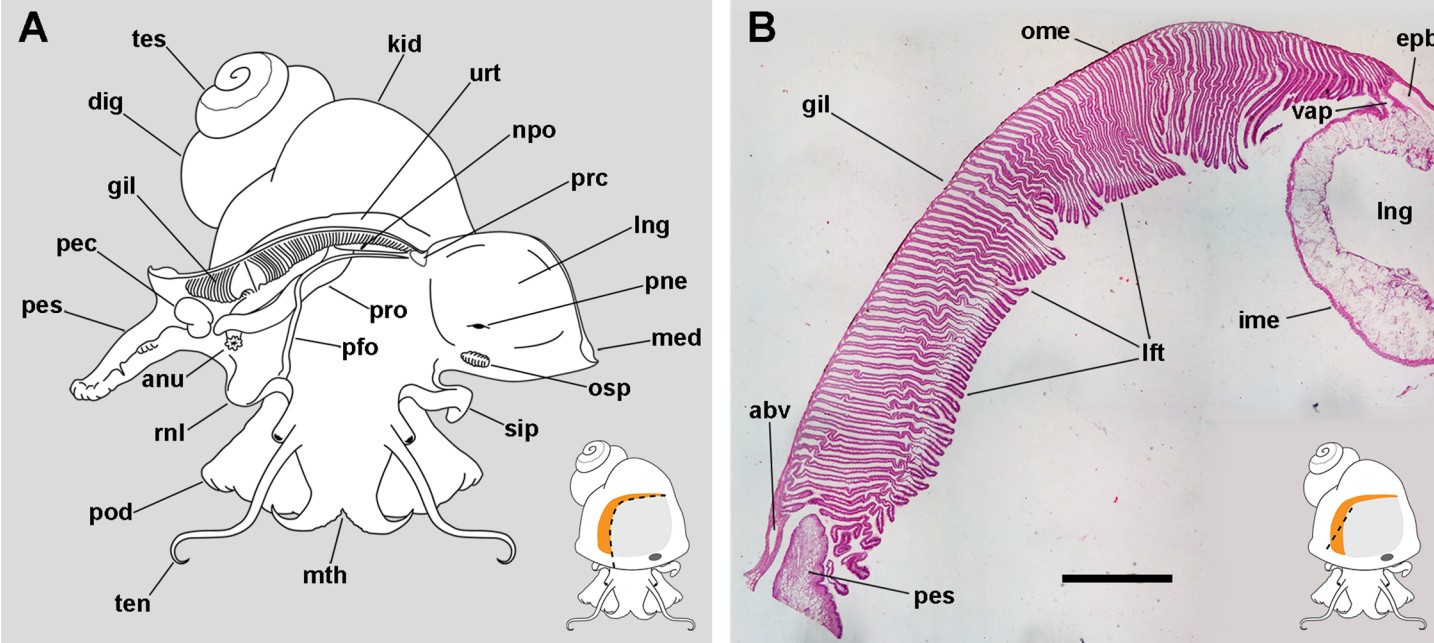

**Figure 1 The gill and the pallial complex.** (A) Diagram of the pallial cavity of a male animal, opened as indicated in the thumbnail sketch at the right lower corner. (B) Panoramic section of the single row of rather parallel leaflets of the monopectinate gill that hang from the gill's base, below the outer mantle; the approximate position of this section is indicated in the thumbnail sketch at the right lower corner. Haematoxylin–eosin. Scale bar represents 1 mm. Abbreviations: abv, afferent branchial vessel; anu, anus; gil, gill; dig, digestive gland; epb, efferent pulmobranchial vessel; ime, inner mantle epithelium; kid, kidney; lft, gill leaflets; lng, lung; med, mantle edge; mth, mouth; npo, nephropore; ome, outer mantle epithelium; osp, osphradium; pec, penial complex; pes, penial sheath; pfo, pallial fold; pne, pneumostome; pod, foot; rnl, right nuchal lobe; prc, pericardium; pro, prostate; sip, siphon; ten, tentacle; tes, testis; urt, ureter; vap, ventral afferent pulmonary vessel.

the *rectal* and *right pallial sinuses* that drain blood from the visceral hump. These afferents join the *afferent branchial vessel*, which continues as the *afferent pulmonary vessel*. Blood from the *afferent branchial vessel* flows through the gill leaflets to the *efferent pulmobranchial vessel* and then to the heart auricle, or alternatively, to the *ventral afferent pulmonary vessel* that irrigates the right half of the lung floor, thus integrating branchial and pulmonary circulation.

The haemocoel within each gill leaflet (Figs. 2D–2F; Fig. S2) extends as a lamina interrupted by trabeculae connecting both lateral surfaces of the leaflet, and is identified here as the *laminar leaflet sinus*. However, there are also two rather continuous haemocoelic sinuses in each leaflet: a *marginal leaflet sinus* runs along the free border of each leaflet, while the other, namely the *basal leaflet sinus*, extends at the base of the gill as a short cut between the *afferent branchial vessel* and the *efferent pulmobranchial vessel*. These sinuses communicate extensively with the *laminar leaflet sinus*, which also connects with the *ventral afferent pulmonary vessel*.

## Branchial nerves and their origins
Innervation of the gill of *Pomacea canaliculata* comes from the *supraoesophageal ganglion* and an *accessory visceral ganglion*, which are diagrammatically shown in Fig. 3A.

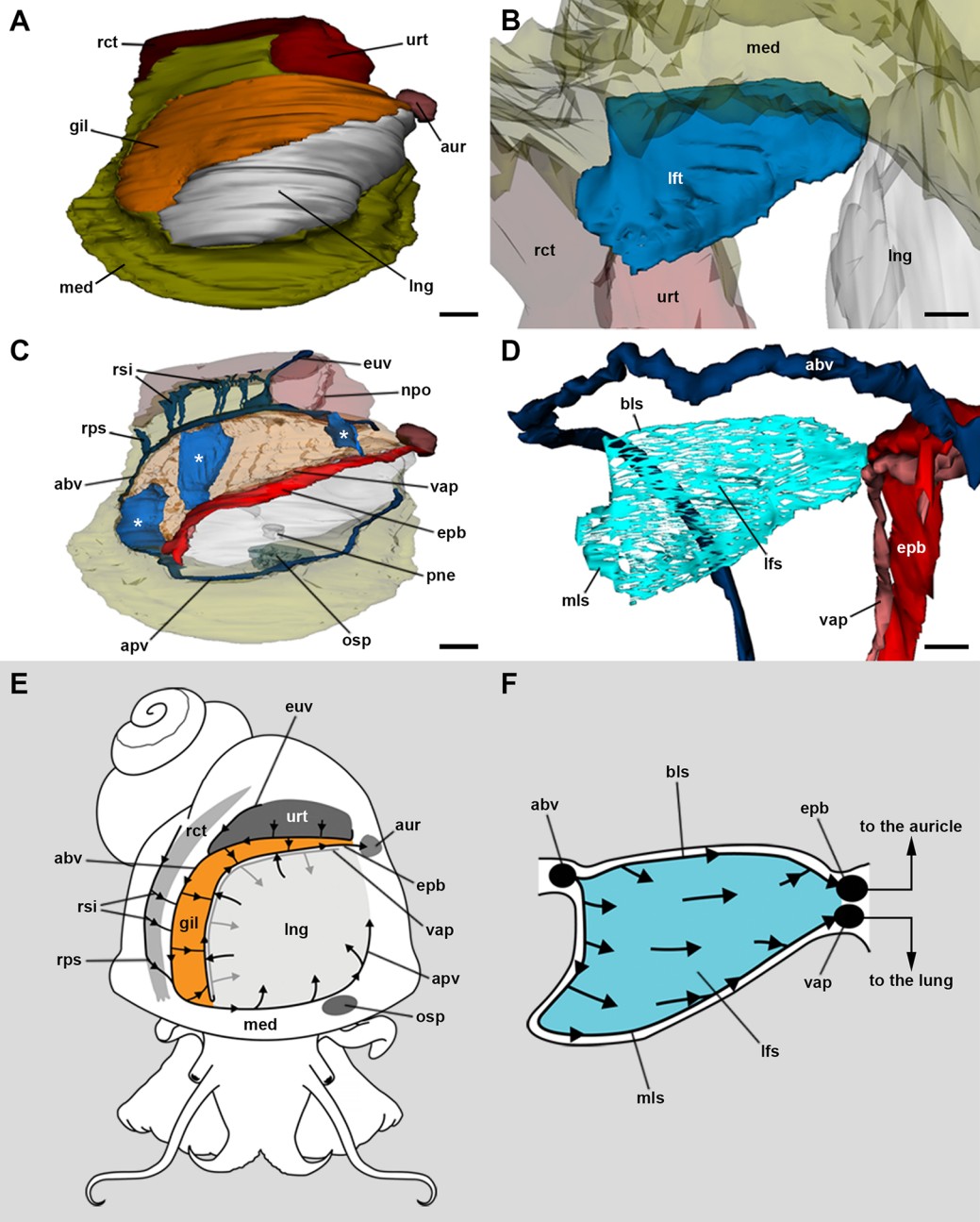

**Figure 2 Computerised 3D rendering of the blood system of the gill.** (A) Dorsal view of the pallial complex. The gill (orange) occupies the right and posterior portion of the roof of the pallial cavity, bordering the lung (which is collapsed, see Methods) and the ureter. (B) Lateral view of a single gill leaflet. (C) The gill's blood supply; three gill leaflets are indicated by asterisks. (D) Blood sinuses in a single gill leaflet. (E) Diagram of the proposed blood flow to and from the gill. (F) Diagram of proposed blood flow within a gill leaflet. Scale bars represent: (A) and (C) 1 mm; (B) and (D) 500 μm. Abbreviations: abv, afferent branchial vessel; apv, afferent pulmonary vessel; aur, auricle; bls, basal leaflet sinus; gil, gill; epb, efferent pulmobranchial vessel; euv, efferent ureteral vessel (=efferent renal vein); lfs, laminar leaflet sinus; lft, gill leaflet; lng, lung; med, mantle edge; mls, marginal leaflet sinus; npo, nephropore; osp, osphradium; pne, pneumostome; rct, rectum; rps, right pallial sinus; rsi, rectal sinus; urt, ureter; vap, ventral afferent pulmonary vessel.

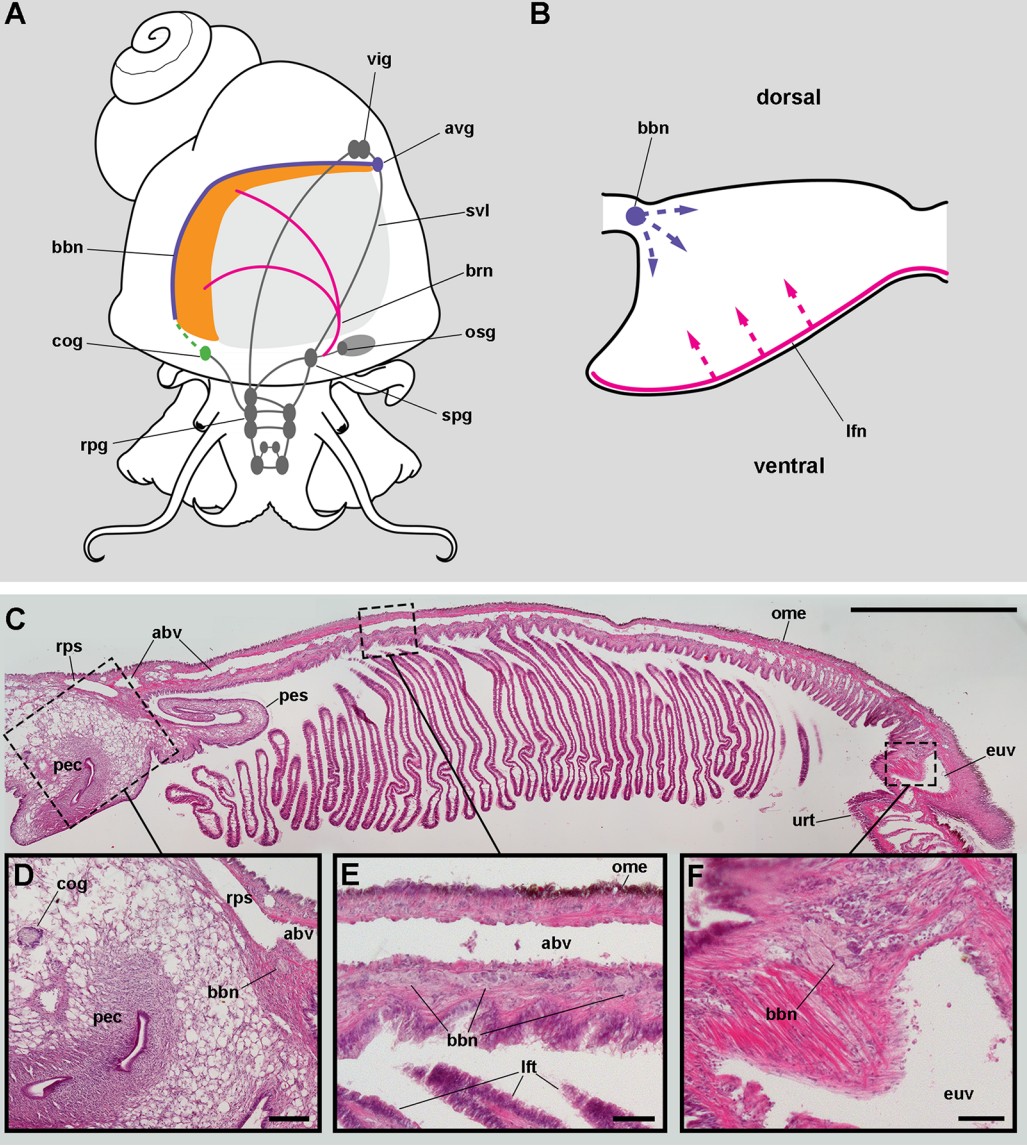

**Figure 3 Gill innervation.** (A) Diagram showing the major ganglia (grey), the nerves, and the accessory ganglia supplying branchial innervation. The gill is innervated from the supraoesophageal ganglion by branches of the branchial nerve (pink). The branchial base nerve originates in the accessory visceral ganglion (violet). The copulatory ganglion may also contribute to branchial innervation (green). (B) Diagram of nerves within each gill leaflet and presumptive origin of the fine innervation (dashed lines). (C) Panoramic section of the gill, showing the branchial base nerve lying alongside the afferent branchial vessel. (D) Detail of the penial complex, showing the copulatory ganglion in the proximity of the branchial base nerve. (E) Detail of the gill base showing the branchial base nerve and the afferent branchial vessel. (F) Detail showing the branchial base nerve in the proximity of the efferent ureteral vessel. Haematoxylin–eosin; parts (D–F) correspond to sections adjacent to that in part (C). Scale bars represent: (C) 1 cm; (D) 1 mm; (E–F) 500 μm. Abbreviations: abv, afferent branchial vessel; avg, accessory visceral ganglion; bbn, branchial base nerve; brn, branchial nerve; cog, copulatory ganglion; euv, efferent ureteral vessel (=efferent renal vein); lfn, leaflet nerve; lft, gill leaflet; ome, outer mantle epithelium; pec, penial complex; pes, penial sheath; rpg, right pleural ganglion; rps, right pallial sinus; spg, supraoesophageal ganglion; svl, supraoesophageal portion of the visceral loop; urt, ureter; vig, visceral ganglion.

Nerve branches of the *branchial nerve* arising from the *supraoesophageal ganglion* go through the lung roof (perhaps giving off neurites that innervate the lung roof) and end at the base of the gill leaflets (C Rodriguez, GI Prieto, IA Vega & A Castro-Vazquez, 2019, unpublished data). Each gill leaflet, however, shows a nerve along its efferent border (Fig. 3B). These *leaflet nerves* presumably originate from branches of the *branchial nerve*.

The *accessory visceral ganglion* is located along the supraoesophageal portion of the visceral loop, close to the pericardium, and gives off a nerve longitudinally traversing the ureter and running along the base of the gill, next to the *afferent branchial vessel* (Fig. 3C). The latter nerve has not been described for *Pomacea canaliculata* (or for any other ampullariid) and it is here referred to as the *branchial base nerve*, which accompanies the *afferent branchial vessel*.

Also, neurite bundles arising from the *copulatory ganglion* are likely to join the *branchial base nerve* through its anterior end (Fig. 3D), at least in male animals. The *copulatory ganglion* lies close to the anterior end of the gill, in the proximity of the anus and the copulatory apparatus.

## The gill leaflets and their regions

Four regions may be distinguished in each leaflet that are characterised by different epithelia (Figs. 4A and 5A). Under scanning electron microscopy, region I appears covered by microvillar cells and interspersed ciliary cells (Fig. 5A), whereas regions III and IV are respectively covered with cells bearing either long (Fig. 5B) or short cilia (Fig. 5C). Region II differs from region I in that it shows no ciliary cells (Figs. 4D and 4E). A summary of data is provided in Table 1.

The *laminar leaflet sinus* occupies the central space of each leaflet and is bordered by a thin fibromuscular layer, which underlies the epithelium (Fig. 4). The sinus is traversed by trabeculae (Figs. 4B and 4C), which in regions I and II are thinner than in region III (Figs. 4D and 4E).

The *leaflet nerve* (Fig. 6A) runs along the efferent *marginal leaflet sinus* and it is partly sheathed in a bundle of longitudinal muscle fibres, which is U-shaped in sections (Fig. 6A) and appears as a distinct structure under scanning electron microscopy (Fig. 6B). The *leaflet nerve* is composed of tightly packed neurites and some glial processes (Figs. 6C and 6D), whereas the muscular bundle is composed of thick fibres in a dense collagen matrix (Fig. 6E).

## Epithelial cell types

The epithelium varies widely in the different regions of each leaflet (data is summarised in Table 2). In region I, it is columnar or low columnar (20–40 µm), and it is mainly composed of either clear or dark microvillar, mitochondria-rich cells, hereafter referred to as α-cells and β-cells, respectively (Fig. 7). Besides that, there are also interspersed ciliary cells and a few secretory cells. These cells (identified as C1, S1 and S2 cells, respectively) will be described for region IV, where they are more abundant.

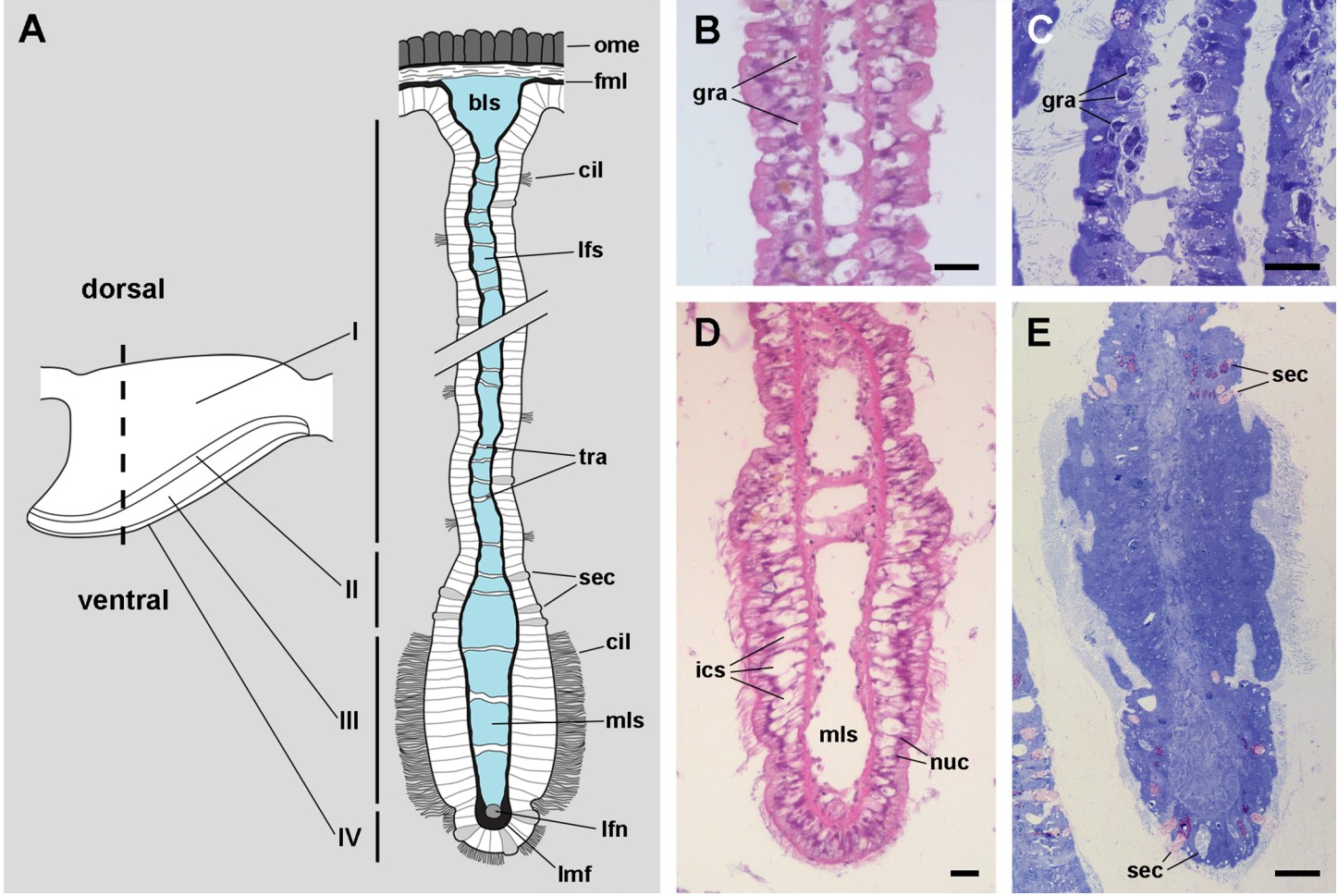

**Figure 4  The gill leaflets and their regions (light microscopy).** (A) Diagram of the four regions of the gill leaflets, which differ in the cell types of its covering epithelium and underlying structures. (B–C) Region I occupies the largest part of the leaflet, while regions II–IV. (D–E) constitute its thickened margin. Scale bars represent 20 μm. Haematoxylin-eosin or toluidine blue. Abbreviations: bls, basal leaflet sinus; cil, cilia; fml, fibro-muscular layer; gra, granulocytes; ics, intercellular spaces; lfn, leaflet nerve; lfs, laminar leaflet sinus; lmf, bundle of longitudinal muscle fibres; mls, marginal leaflet sinus; nuc, epithelial cell nuclei; ome, outer mantle epithelium; sec, secretory cells; tra, trabeculae.

Alpha-cells are characterised by euchromatic nuclei and conspicuous nucleoli. The cytoplasm contains numerous long mitochondria with well-defined cristae and glycogen deposits (Figs. 8A–8C). Apically, these cells show few and rather short microvilli, and underlying membrane-bound bundles of electron-dense filaments/tubules. There is also a well-developed endomembrane vesicular system, as well as multivesicular (Fig. 8B) and multilamellar bodies (Fig. 8C).

Contrasting with α-cells, β-cells bear heterochromatic nuclei and the cytoplasm with numerous and tightly packed mitochondria (Figs. 8D–8F). The surface area of the apical domain is increased by numerous and ramified microvilli (see Fig. 5A). An apical narrow band of homogeneous cytoplasm is seen below the microvilli, together with the apical ends of an extensive tubular system, which extends to the underlying mitochondrial conglomerate (Fig. 8E). Multivesicular bodies as well as presumptively

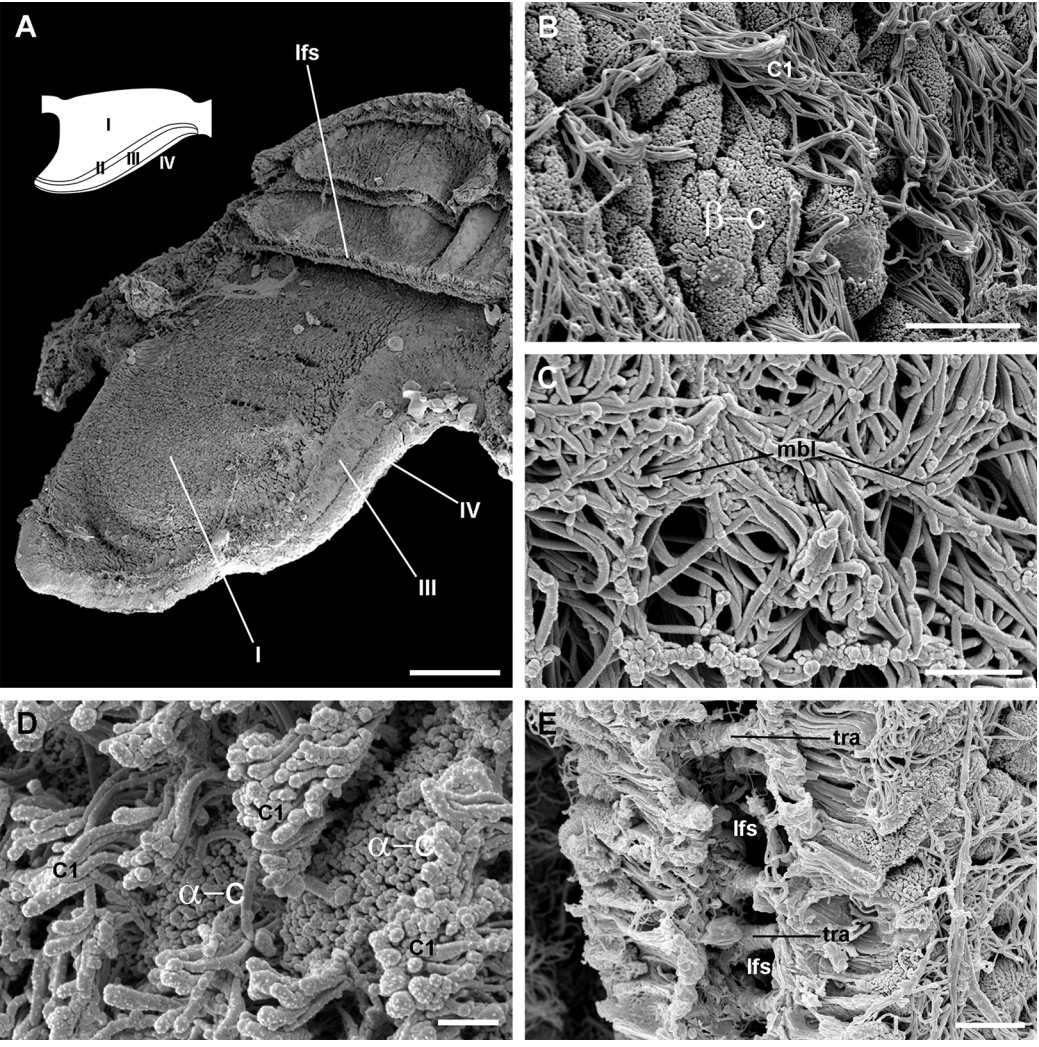

**Figure 5 Apical specialisations on the gill surface (scanning electron microscopy).** (A) Three adjacent gill leaflets, two of them sectioned to show the laminar leaflet sinus. Regions described in Fig. 4A are indicated in the thumbnail sketch. (B) Region I exhibiting the cilia of α-cells and the ramified microvilli of β-cells. (C) Region III exhibiting the long cilia of C2 cells with the characteristic membrane blebs. (D) Region IV exhibiting bundles of the short cilia of C1 cells, and interspersed spaces showing the microvilli of α-cells. (E) A cut through region I of a gill leaflet, showing the laminar leaflet sinus, the trabeculae traversing it, and the covering leaflet epithelia showing α-cells, β-cells and C1 cells. Scale bars represent: (A) 200 μm; (B) 10 μm; (C–E) 2 μm. Abbreviations: α-c, α-cells; β-c, β-cells; C1, short cilia cells; lfs, laminar leaflet sinus; mbl, membrane blebs on the long cilia; tra, trabeculae.

degenerative bodies, such as myeloid bodies and fibrogranular material, are also abundant in the perinuclear region of these cells (Fig. 8F). Alpha-cells and β-cells alternate in approximately equal numbers. Numerous granulocytes lay inside epithelial intercellular spaces all along region I (Fig. 9), although they may also occur in other regions (see Section 'Apical intercellular junctions and the basolateral domain of epithelial cells').

A single cell type constitutes the epithelium in region III, which rests on a well-defined and electron-dense basal lamina (Fig. 10). It consists of cells with slender nuclei, very

**Table 1 Cell types and other features of the gill leaflet regions.**

| Region | Epithelial cell types | | | Epithelial intercellular spaces | Underlying tissues |
|---|---|---|---|---|---|
| | Secretory cells | Microvillar cells | Ciliary cells | | |
| I | α and β | C1 | S1 and S2 (scarce) | Extensive spaces, with numerous granulocytes | Thin basal lamina<br>Loose fibromuscular tissue<br>Thin trabeculae cross the laminar leaflet sinus |
| II | α | None | S1 and S2 (abundant) | Narrow spaces, scarce granulocytes | Thick basal lamina<br>Dense fibromuscular tissue<br>Thin trabeculae cross the marginal leaflet sinus |
| III | None | C2 | None | Extensive spaces, scarce granulocytes | Thick basal lamina<br>Dense fibromuscular tissue<br>Thick trabeculae cross the marginal leaflet sinus |
| IV | α | C1 | Abundant S1 and S2 | Narrow spaces, scarce granulocytes | Thick basal lamina<br>Muscular bundle<br>Leaflet nerve |

**Note:**
α, α-cells; β, β-cells; C1, short cilia cells; C2, long cilia cells; S1, metachromatic secretory cells; S2, orthochromatic secretory cells.

long cilia and short microvilli (C2 cells), and whose basolateral plasma membranes often enclose extensive and apparently dynamic intercellular spaces. The cytoplasm contains abundant rough endoplasmic reticulum and glycogen deposits (Figs. 10A and 10B). Transverse sections of cilia (Fig. 10C) show the typical 9 + 2 microtubule arrangement as well as membrane blebs (Fig. 10C, arrows). As α-cells, these cells have membrane-bound bundles of electron-dense filaments/tubules in the subapical region (Fig. 10D).

The epithelium in region IV rests on a well-defined and electron-dense basal lamina and exhibits the same cell types as epithelium in region I, except for β-cells (Fig. 11A). However, ciliary C1 cells are the most abundant cell type here. They are characterised by a heterochromatic nucleus and an electron-dense cytoplasm that contains large dense-cored granules and glycogen deposits (Fig. 11B). The subapical region is similar to those of α-cells, but apically there are finger-like microvilli and short cilia (Fig. 11B). Secretory S1 cells are mucous cells with merging granules containing an electron-dense mesh above the nucleus (Fig. 11C), and granules with a looser electron-dense mesh in the apical domain (Fig. 11D). These granules would correspond to the dense and loose metachromatic accumulations as seen in toluidine blue preparations (see Figs. 4E and 6A). On the other hand, secretory S2 cells have their cytoplasm almost filled with granules with a low electron-dense core (Fig. 11E), which are orthochromatic in toluidine blue preparations (Figs. 4E and 6A).

Epithelium in region II exhibits α-cells, and abundant S1 and S2 cells (Fig. 4). The epithelia in regions III and IV are those directly exposed to the incoming water current that flows over the gill, and both the long and short cilia may contribute to the water movement.

## Apical intercellular junctions and the basolateral domain of epithelial cells

All epithelia examined showed two types of cellular junctions in an orderly fashion. Apical adherent junctions (also called *zonula adhaerens* or 'desmosome belt'; Fig. 12A)

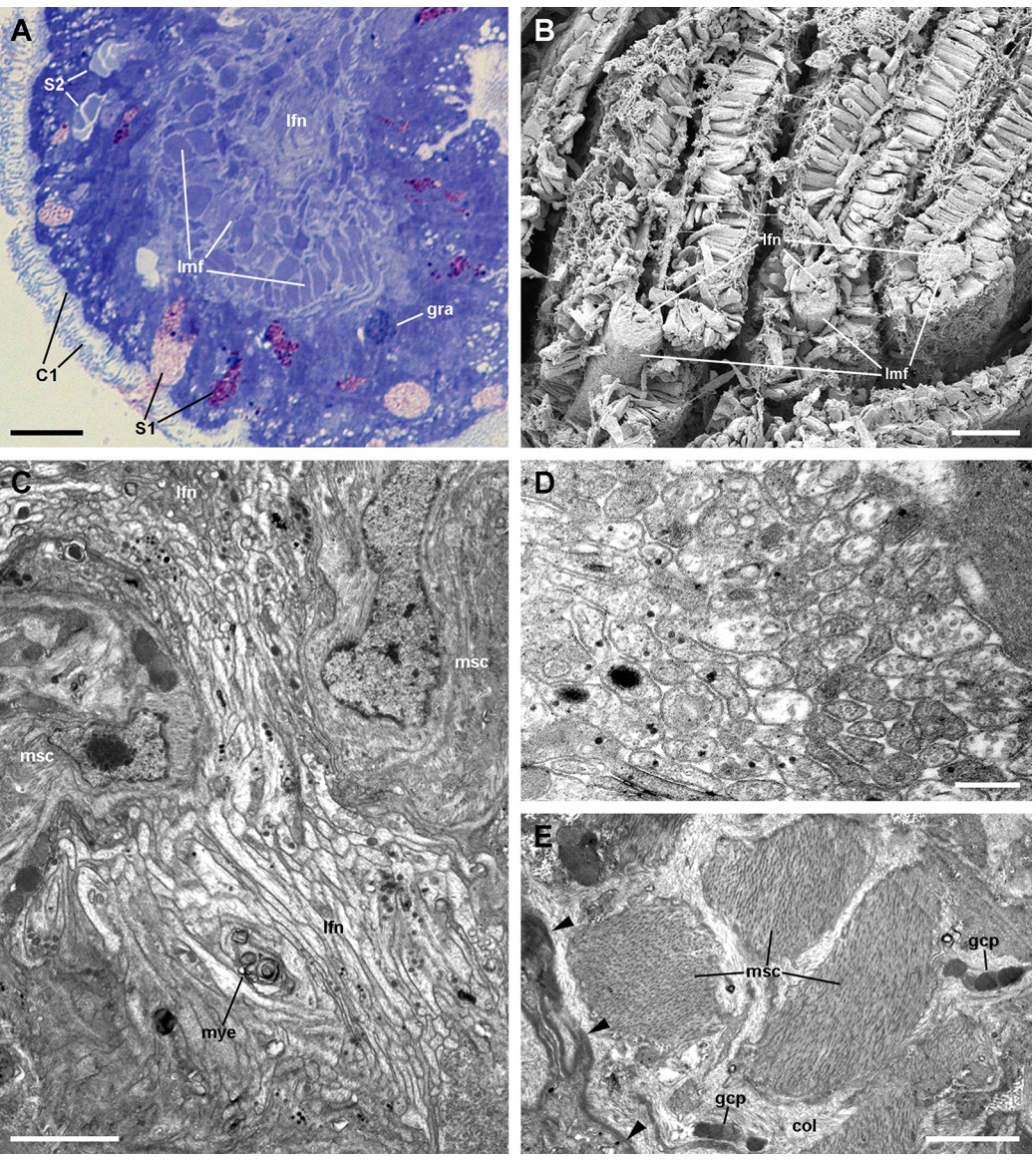

**Figure 6 The muscular bundle and the leaflet nerve (light and electron microscopy).** (A) Margin of a gill leaflet showing the covering epithelium, the muscular bundle and the leaflet nerve. Toluidine blue. (B) Razor blade cuts of three gill leaflets showing the muscular bundles as well-defined units beneath the covering epithelium. Scanning electron microscopy. (C) Tangential section of the leaflet border showing two muscular cells pertaining to the muscular bundle and a fairly longitudinal section of the leaflet nerve. Transmission electron microscopy. (D) A high magnification of the leaflet nerve showing tightly packed neurites containing neurotubules, clear vesicles or electron-dense granules of different sizes. (E) A section through the muscular bundle showing large muscle fibres containing myofibrils, and that are embedded in a collagen matrix where glial processes are found. Arrowheads indicate the basal lamina of the covering epithelium. Scale bars represent: (A) 10 μm; (B) 25 μm; (C–D) 1 μm; (E) 250 nm. Abbreviations: C1, short cilia cells; col, collagen matrix; gcp, glial cell process; gra, granulocyte; lfn, leaflet nerve; lmf, bundle of longitudinal muscle fibres; msc, muscle cell; mye, myeloid body; S1, metachromatic secretory cells; S2, orthochromatic secretory cells.

are followed by septate junctions (Fig. 12B), which are often interrupted by intercellular canaliculi (Figs. 12A and 12B). Rather frequently, the canaliculi contain a globular, unidentified material (Fig. 12A). Below the septate junctions, the adjacent plasma

membranes are separated by larger spaces, which increase in size towards the basal domain (Fig. 12C).

The basolateral domain of epithelial cells in all regions of a leaflet is a labyrinth of intermingled thin extensions that project towards the underlying connective tissue. There are conspicuous spaces between epithelial cells, which are frequently occupied by granulocytes (Fig. 9). Small neurite bundles, with or without accompanying glial cells, are also found in the intercellular spaces, thus providing direct intraepithelial innervation (Fig. 9A, arrows). A collagen matrix and sparse muscle fibres form the underlying connective tissue, which delimits the leaflet haemocoel. Neurite bundles accompanied by glial cells are frequently found in this tissue (see Fig. 9A), and these will be further described in the Section 'Fibromuscular tissue and fine innervation'.

A notable feature of this epithelium is the occurrence of granulocytes, with eccentric nuclei and conspicuous nucleoli, electron-dense granules, and areas of glycogen deposits. Granulocytes are often fully enclosed into expanded intercellular spaces (Figs. 9A and 9B), but sometimes, discontinuities in the mesh of basal projections of epithelial cells are found (Fig. 9C). These discontinuities connect the intercellular spaces directly with the basal lamina, which is evident because of its well-delimited *lamina densa* with interspersed electron-dense thickenings (Fig. 9C, arrowheads).

## Fibromuscular tissue and fine innervation

Underlying the basal lamina there are muscle cells embedded in a collagen matrix (Fig. 13A). These muscle cells show myofibrils and scarce mitochondria. The cell surface shows numerous electron-dense anchoring junctions composed of an external brush-like plaque and an internal amorphous but electron-dense layer (Fig. 13B). The fibrils of the brush-like plaques are often continuous with those of other cell plaques or with thickenings of the *lamina densa* (Figs. 13A and 13B, arrowheads). These peculiar structures lie over clear cytoplasmic areas, which are traversed by thin cytoskeletal fibres.

Numerous neurite bundles that traverse the connective tissue and sometimes go into the epithelial intercellular spaces provide leaflet innervation. Neurites form bundles that are flanked by glial cell processes (Fig. 13C). Glial cells, or less frequently, uncovered neurites are in contact with muscle cells. Glial cells have rounded electron-dense, membrane-bound, granules of two different sizes (either ~400 or ~100 nm wide). Neurites have an electron-lucent axoplasm with neurofilaments. Fibre enlargements (presumptive nerve endings) contain ~80 nm wide granules of variable electron density and ~50 nm wide clear vesicles (Fig. 13C).

## DISCUSSION

### Comparative aspects of the gill leaflets and supporting structures

Most gastropod ctenidia are composed of triangular leaflets (*Lindberg & Ponder, 2001*) that commonly have four ciliary fields: lateral, frontal, abfrontal and terminal. Lateral cilia are placed in two rows alongside the efferent border of the leaflet and provide ventilation, while the others cover the free edges of the leaflet and are engaged in cleansing (*Lindberg & Ponder, 2001*; *Yonge, 1947*). The gill leaflets of *Pomacea canaliculata* are rather

**Table 2 Features of the cell types in the gill epithelium.**

| Cell type | Apical specialisations | Nucleus and cytoplasm | Endomembrane system | Other membrane-bound bodies |
|---|---|---|---|---|
| α | Few and short, finger-like microvilli | Euchromatic nucleus Abundant, long mitochondria | Abundant RER Golgi bodies Vesicular system Multivesicular bodies Multilamellar bodies | Bundles of electron-dense tubules/filaments Dense-cored granules (in region IV only) |
| β | Numerous and long, ramified microvilli | Heterochromatic nucleus Tightly-packed, short mitochondria | Tubular system Multivesicular bodies Myeloid and fibrogranular bodies | Few and small bundles of electron-dense tubules/filaments |
| C1 | Short cilia with membrane blebs Short, finger-like microvilli | Heterochromatic nucleus Rather dark cytoplasm | Vesicular system Multivesicular bodies | Abundant and large dense-cored granules Bundles of electron-dense tubules/filaments |
| C2 | Very long cilia with membrane blebs Short, finger-like microvilli | Euchromatic nucleus Rather dark cytoplasm | Abundant RER | Bundles of electron-dense tubules/filaments |
| S1 | None | Heterochromatic nucleus Rather dark cytoplasm | Abundant RER Golgi bodies | Mucinogen granules (basally, with an electron-dense mesh; apically, with a looser electron-dense mesh) |
| S2 | None | Euchromatic nucleus Clear cytoplasm | Abundant RER Golgi bodies | Granules with moderately electron-dense cores |
| G | – | Euchromatic nucleus Clear cytoplasm | Golgi bodies | R granules |

**Note:**
α, α-cells; β, β-cells; C1, short cilia cells; C2, long cilia cells; S1, metachromatic secretory cells; S2, orthochromatic secretory cells; G, granulocytes; RER, rough endoplasmic reticulum.

triangular and have well-developed lateral and frontal ciliary fields, which are here referred to as regions III and IV respectively (Fig. 4), but have no abfrontal or terminal cilia. The identity and disposition of the ciliary fields, as well as the rather small variation in shape and size of the leaflets along the gill in this species (Fig. S1), are similar to those of *M. cornuarietis* (Demian, 1965; Lutfy & Demian, 1965).

An efficient respiratory organ requires both a large and thin surface area. This respiratory surface should be fully exposed to the external medium to allow gas exchange (Maina & West, 2005). Therefore, some mechanical rigidity to support a deployed gill is indispensable to permit adequate gas exchange. Structures that support the weight of gills have been found in different taxa, such as the cartilaginous or bony supporting rods (gill rays), the interbranchial septa and the 'pillar cells' in fishes (Evans, Piermarini & Choe, 2005), the cuticle, the intralamellar septa, and the pillar cells in crustaceans (Farrelly & Greenaway, 1987, 1992), and the 'supporting skeletal rods' and 'trabecular cells' in some molluscs (Eertman, 1996; Hyman, 1967; Nakao, 1975; Ridewood, 1903; Wanichanon et al., 2004; Yonge, 1947).

A 'skeletal rod' has been mentioned several times for the Ampullariidae (Berthold, 1991; Haszprunar, 1988; Ponder & Lindberg, 1997; Salvini-Plawen & Haszprunar,

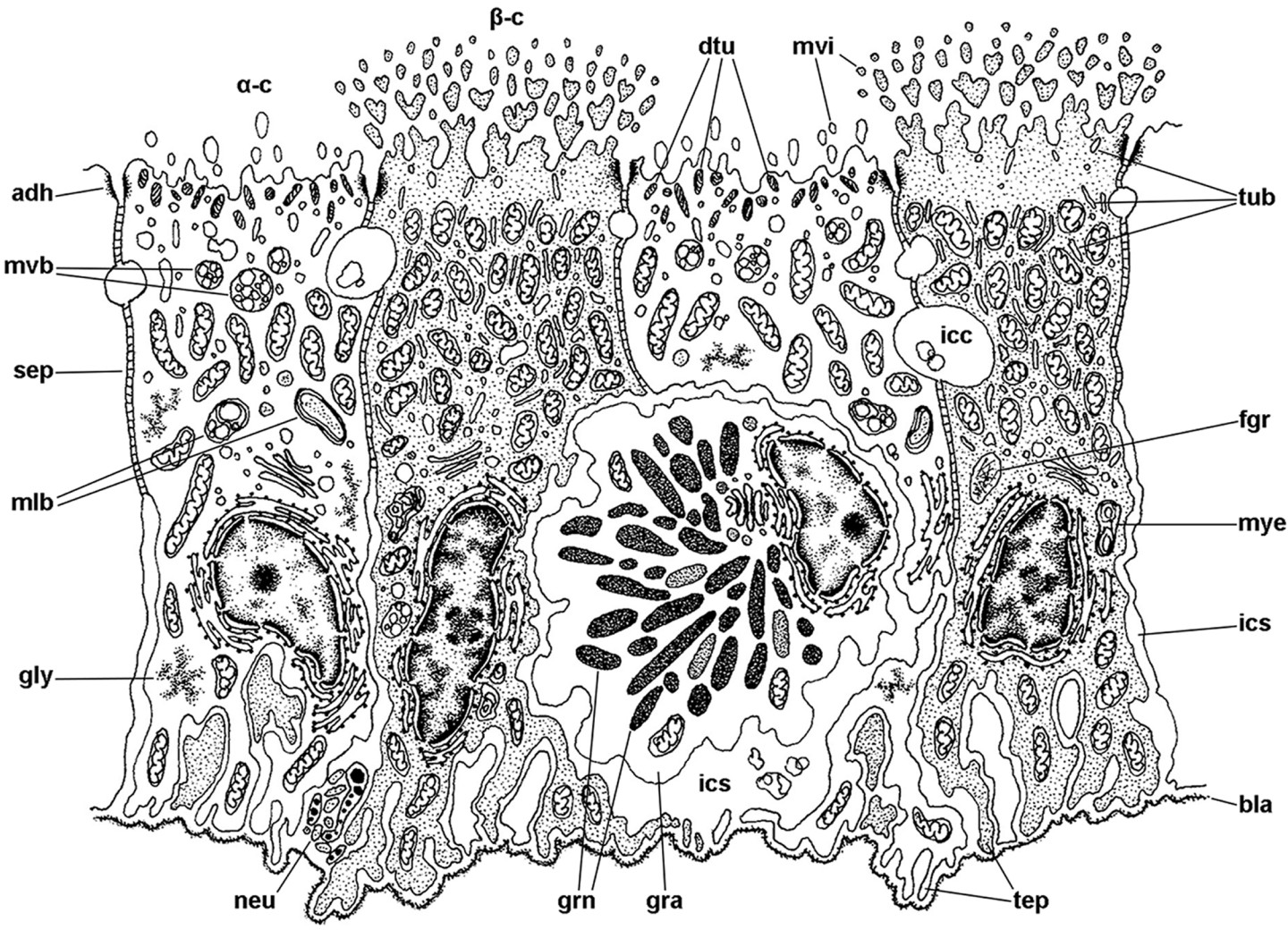

**Figure 7 Mitochondria-rich cells in region I of the gill leaflets (diagram).** The two main cell types found in region I are α- and β-cells. In addition, granulocytes occur within epithelial intercellular spaces. Abbreviations: adh, adherent junction; α-c, alpha-cell; β-c, beta-cell; bla, basal lamina; dtu, bundles of electron-dense tubules; gra, granulocyte; grn, R granules; fgr, fibrogranular material; gly, glycogen deposit; icc, intercellular canaliculi; ics, intercellular space; mlb, multilamellar bodies; mvb, multivesicular bodies; mvi, microvilli; mye, myeloid bodies; neu, neurite bundle; sep, septate junction; tep, thin epithelial projections; tub, tubular system.             

*1987*), but no information on its microstructure or composition has been given. In *Pomacea canaliculata*, we found a bundle of thick longitudinal muscle fibres embedded in a dense collagen matrix, which runs along the efferent margin of each gill leaflet (see Fig. 13).Therefore, it is a fibromuscular contractile structure and not a skeletal rod as has been suggested by earlier authors. It accompanies the *leaflet nerve* and the *marginal leaflet sinus*. It is worth noting that likely homologous muscular structures occur in the ampullariids *M. cornuarietis* (*Lutfy & Demian, 1965*) and *Pila globosa* (*Prashad, 1925*). However, neither *Lutfy & Demian (1965)* nor *Prashad (1925)* referred to the existence of a *leaflet nerve*. Considering that the three genera are closely related and that these authors used only light microscopy techniques, which are not best suited to recognise nervous tissue, they perhaps overlooked the *leaflet nerve*.

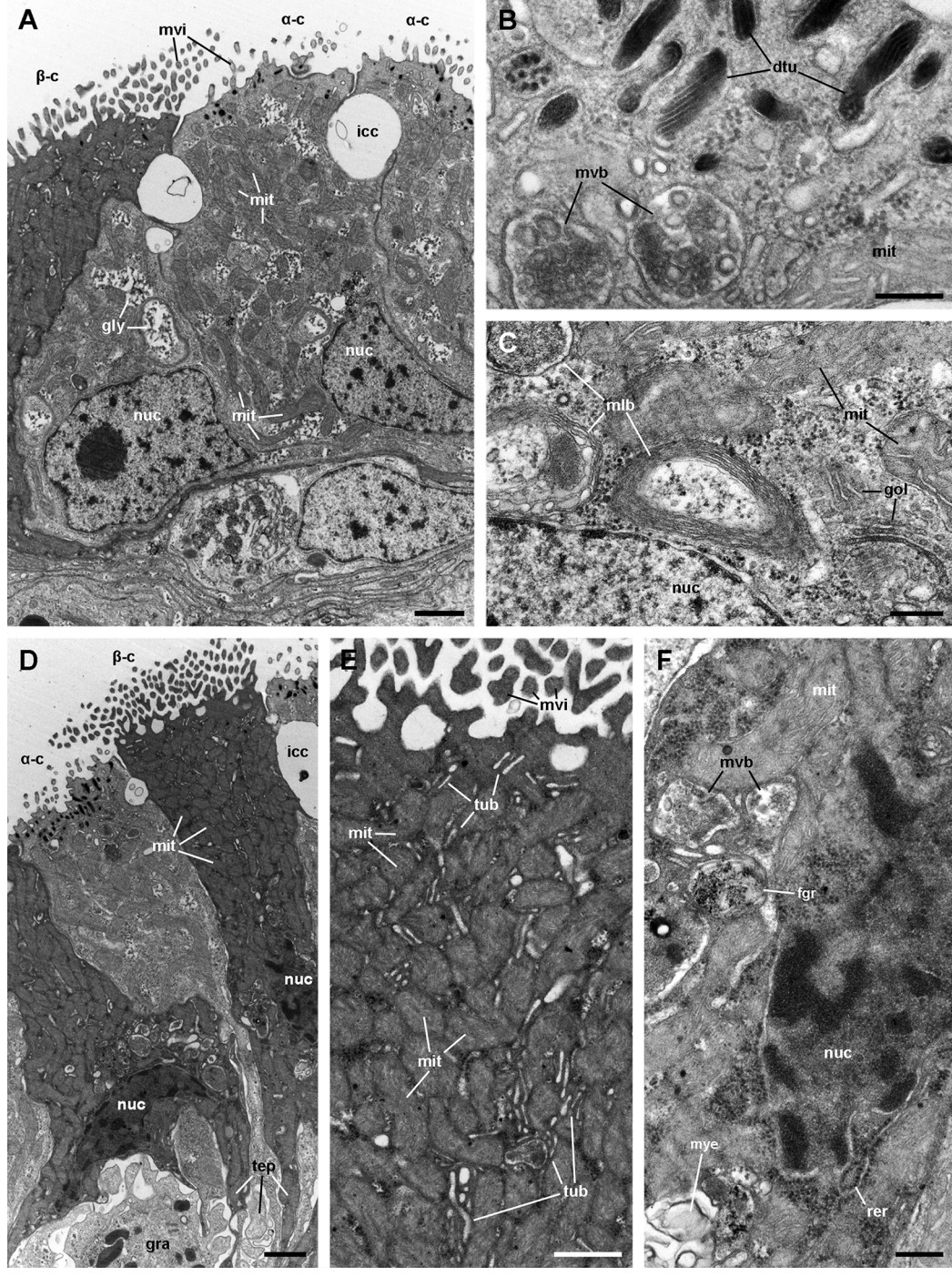

**Figure 8 Mitochondria-rich cells in region I of the gill leaflets (transmission electron microscopy).**
(A) Alpha-cells exhibit few and short microvilli, numerous and long mitochondria with well-defined cristae, and glycogen deposits. Their nuclei are euchromatic with conspicuous nucleoli. (B) Apically, α-cells show numerous membrane-bound bundles of electron-dense tubules/filaments, and a well-developed vesicular system, including multivesicular bodies. (C) Multilamellar bodies and Golgi bodies are found close to the nucleus. (D) Contrasting with α-cells, β-cells have numerous and ramified microvilli. These cells also have numerous tightly-packed mitochondria that fill almost all the cytoplasm. The nuclei are heterochromatic. (E) Beta-cells show an extensive tubular system between the mitochondria. (F) A β-cell showing multivesicular bodies and presumptively degenerative bodies, such as

**Figure 8** (continued)
myeloid bodies and fibrogranular material. Scale bars represent: (A) 1 µm; (B) 200 nm; (C) 250 nm; (D) 1 µm; (E) 500 nm; (F) 250 nm. Abbreviations: α-c, alpha-cell; β-c, beta-cell; dtu, bundles of electron-dense tubules; fgr, fibrogranular material; gly, glycogen deposit; gol, Golgi body; gra, granulocyte; icc, intercellular canaliculi; mit, mitochondria; mlb, multilamellar body; mvb, multivesicular body; mvi, microvilli; mye, myeloid body; nuc, cell nucleus; rer, rough endoplasmic reticulum; tep, thin epithelial projections; tub, tubular-vesicular system.

Trabecular cells are common structures in molluscan gills, whether they are ctenidia (*Gregory, George & McClurg, 1996*; *Le Pennec, Beninger & Herry, 1988*; *Manganaro et al., 2012*; *Nuwayhid, Davies & Elder, 1978*) or secondary gills (*De Villiers & Hodgson, 1987*; *Jonas, 1986*). Trabeculae had been previously observed in ampullariids but were mistakenly considered as septa delimiting parallel blood sinuses (*Andrews, 1965*; *Lutfy & Demian, 1965*; *Prashad, 1925*; *Ranjah, 1942*).

*Nakao (1975)* described in detail the trabecular cells of the bivalve *Anodonta woodiana*. They are modified muscular cells with a cytoplasm almost filled with myofibrils, scarce peripheral organelles, and anchoring junctions. The modified muscular cells underlying the basal lamina in *Pomacea canaliculata* (see Figs. 13A and 13B) show similar characteristics, and thus may correspond to the anchoring part of the trabeculae shown in Figs. 4 and 5C.

## The gill as a respiratory organ

The 3D rendering of the blood system of the gill in *Pomacea canaliculata* suggests that the general pattern of branchial blood circulation is similar to that described by *Andrews (1965)* for the same species; although, there is a major difference that is worth mentioning (other differences can be found in Table S1). The gill leaflets of *Pomacea canaliculata* have a single laminar sinus interrupted by trabeculae and bounded by continuous sinuses at the leaflet borders and base, but with no transverse sinuses as interpreted by *Andrews (1965)* for the same species, or by *Lutfy & Demian (1965)* for *M. cornuarietis*. This laminar arrangement implies a slow sheet-flow of blood, which likely facilitates the exchange of respiratory gases (*Maina, 2000b*, *2002a*) and, perhaps more importantly, of ions. Moreover, this 'sheet-flow design' is in agreement with that found in another molluscan, crustacean, and fish gills (*Booth, 1978*; *Knight & Knight, 1986*; *Maina, 1990*). According to *Andrews (1965)*, ciliary beating should conduct water between the gill leaflets towards a narrow channel bordered by the gill and the pallial fold. An exhalant water stream flows along the course of this channel, thus expelling urine and faeces (*Andrews, 1965*). In this way, a countercurrent mechanism would occur between blood flowing through the leaflets and water flowing between them, facilitating $O_2$ extraction from the water (see Fig. 2F).

However, in spite of having a countercurrent mechanism for $O_2$ extraction, the gas exchange should be hindered by the thickness of the gill epithelium (>20 µm), according to Fick's first law of diffusion (*Maina & West, 2005*). Also, the large number of mitochondria found in epithelial cells (see Figs. 6 and 7) indicates a high oxygen consumption, but this finding contradicts the idea that a respiratory tissue barrier must consume a minimal amount of the oxygen it extracts from the external medium (*Maina, 2000b*). Indeed, a high

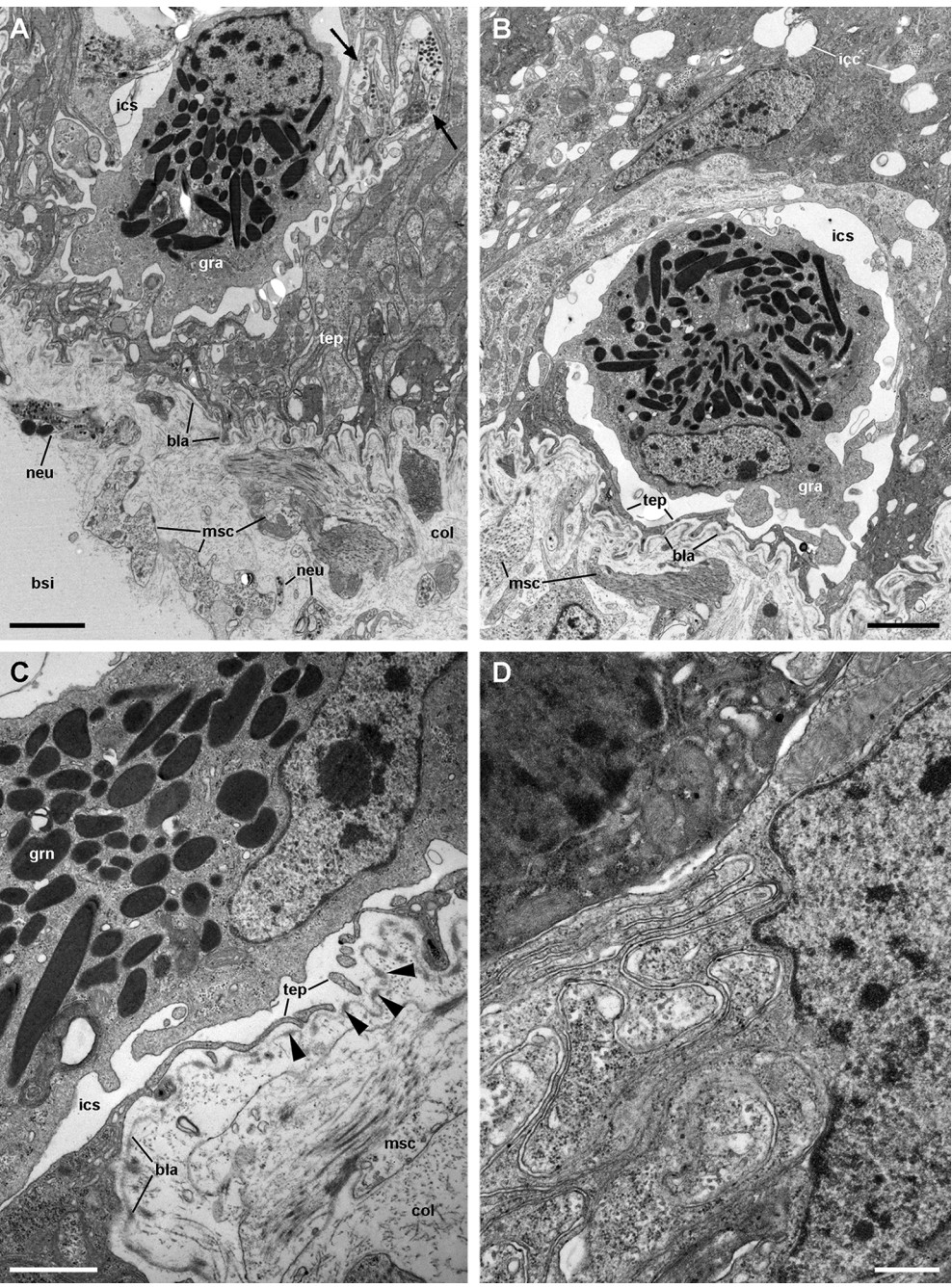

**Figure 9 Structures associated with granulocytes in the basolateral domain of the gill epithelium (transmission electron microscopy).** (A) Labyrinth of thin cellular extensions projecting towards the collagen matrix of the underlying connective tissue, where sparse muscle fibres occur. Subepithelial and intraepithelial (arrows) neurite bundles with accompanying glial cells also occur. (B) A granulocyte in close proximity to the basal lamina occupies an enlarged intercellular space. Small intercellular spaces or canaliculi are also seen. (C) Discontinuities in the basal mesh of epithelial projections connect the inter-cellular spaces directly with the basal lamina, which shows interspersed electron dense thickenings (arrowheads). (D) Basolateral infoldings of an α-cell, adjacent to a β-cell. Scale bars represent: (A–B) 2 μm; (C) 1 μm; (D) 500 nm. Abbreviations: bla, basal lamina; bsi, blood sinus; col, collagen matrix; gra, granulocyte; grn, R granule; icc, intercellular canaliculi; ics, intercellular space; msc, muscle cell; neu, neurite bundle; tep, thin epithelial projections.

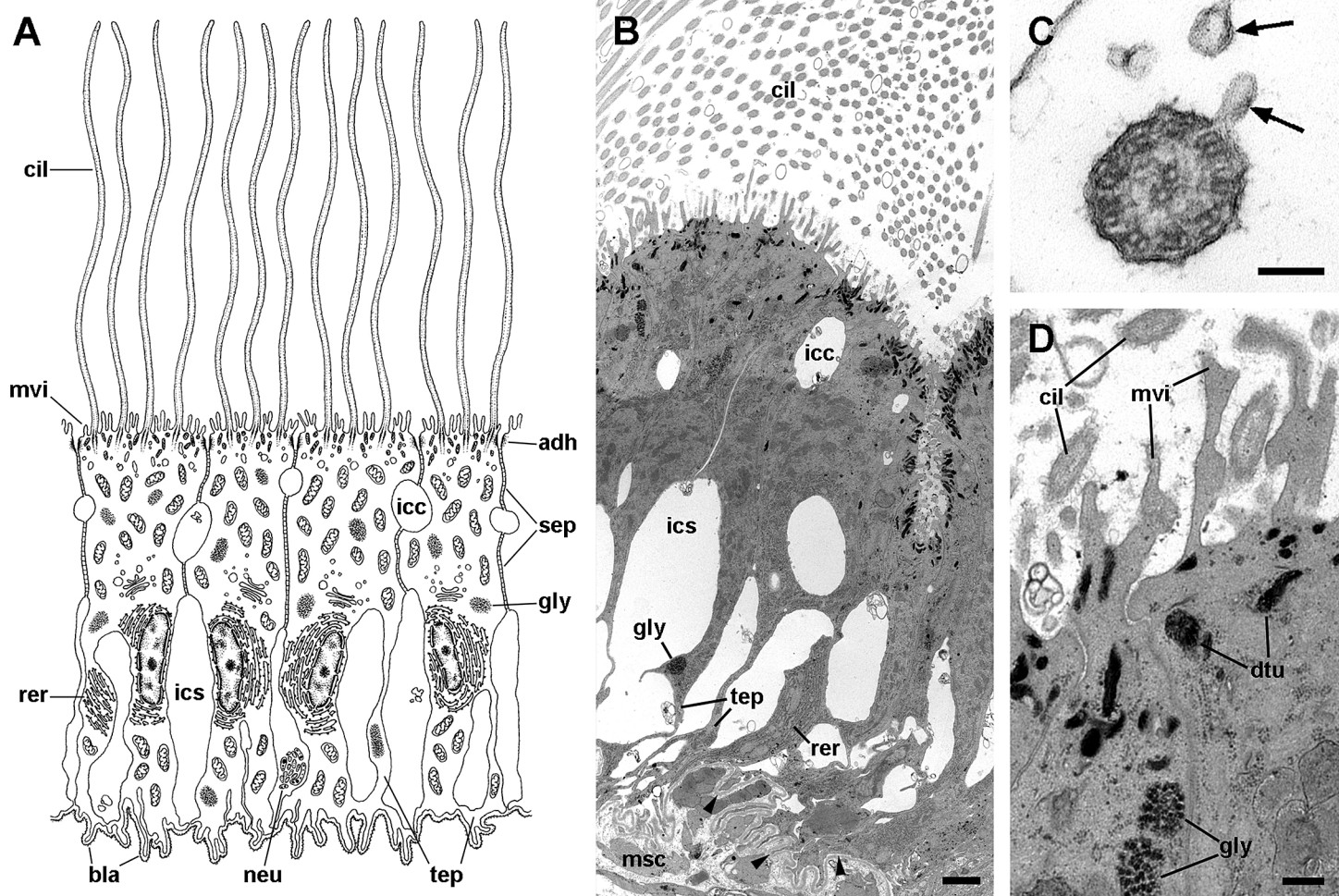

**Figure 10 Long cilia cells (C2) in region III of the gill leaflets (transmission electron microscopy).** (A) Diagram. (B) C2 cells exhibit an electron dense cytoplasm with abundant rough endoplasmic reticulum and glycogen deposits. These cells rest on a thick and electron dense basal lamina (arrowheads). There are extensive intercellular spaces and smaller canaliculi, also seen under light microscopy (Fig. 4D). (C) Transverse section of a cilium shows the typical 9 + 2 microtubule arrangement and membrane blebs (arrows). (D) Membrane-bound bundles of electron-dense tubules/filaments in the apical domain of a C2 cell. Scale bars represent: (B) 1 µm; (C) 50 nm; (D) 250 nm. Abbreviations: adh, adherent junction; bla, basal lamina; cil, cilia; dtu, bundles of electron-dense tubules; gly, glycogen deposit; icc, intercellular canaliculi; ics, intercellular space; msc, muscle cell; mtb, microtubules; mvi, microvilli; neu, neurite bundle; rer, rough endoplasmic reticulum; sep, septate junction; tep, thin epithelial projections.

oxygen consumption rate would be required for ion pumping against concentration gradients, which likely occurs in the gill epithelium. It should be considered, however, that a decrease in septate junctions' length –as occurs in leaflet region III– may shorten the distance between the external and internal media for the passage of molecules via the paracellular pathway (*Yu & Chir, 2017*). It is therefore expected that some downhill diffusion of gases followed this route, which requires no energy expenditure. However, the gill $CO_2$ excretion would be higher than the $O_2$ uptake, because of the higher solubility of $CO_2$ in water. This is true, indeed, for many freshwater bimodal-breathing fishes (*Evans, Piermarini & Choe, 2005*) and crustaceans (*Innes & Taylor, 1986*), in which $CO_2$ excretion occurs mainly through their gills, and $O_2$ uptake occurs mainly through their lungs.

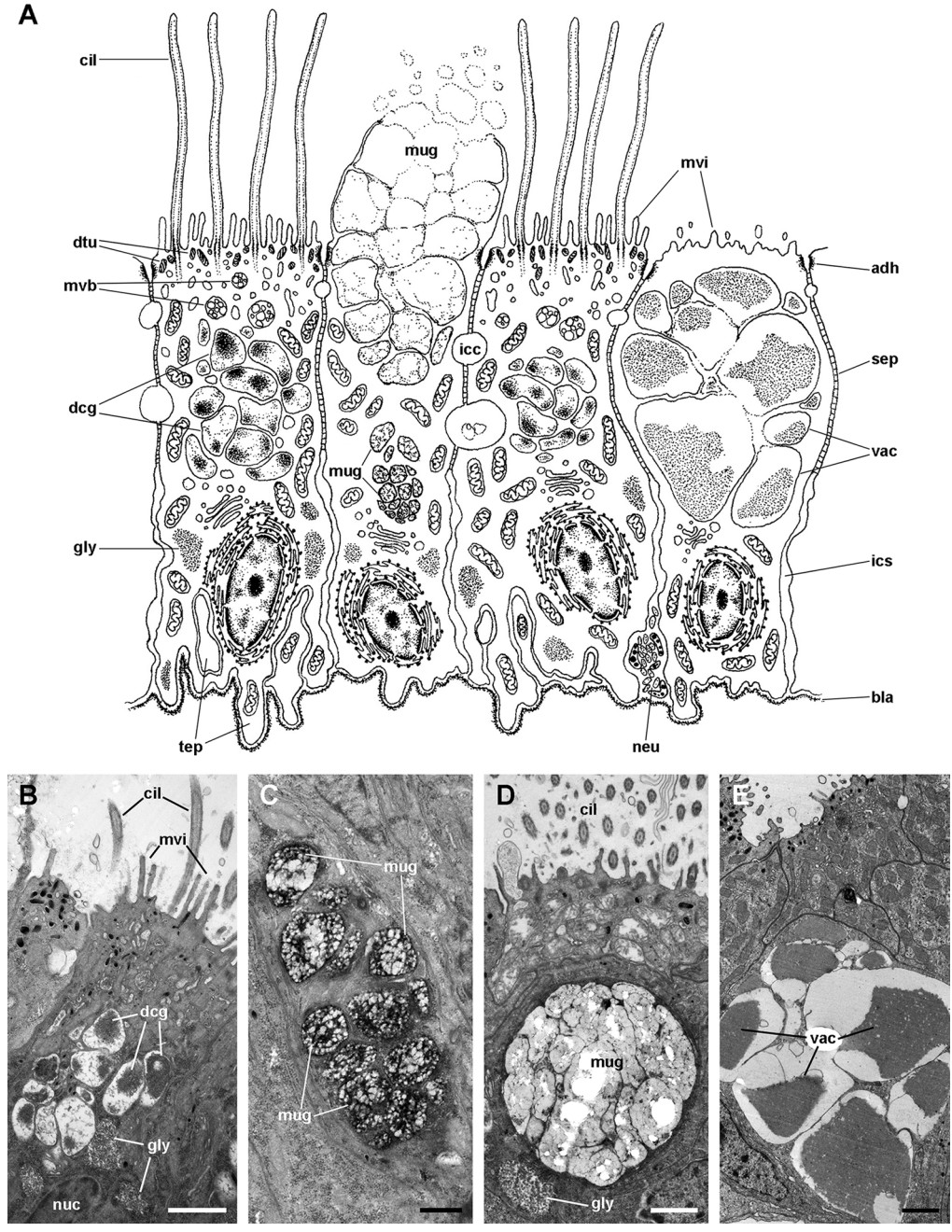

**Figure 11 Ciliary (C1) and secretory (S1 and S2) cells in region IV of the gill leaflets (transmission electron microscopy).** (A) Diagram. (B) C1 cells exhibit a moderately electron-dense cytoplasm containing large dense-cored granules and a heterochromatic nucleus. Apically, there are finger-like microvilli and short cilia with membrane blebs. Glycogen deposits are also found. (C) An S1 cell showing granules above the nucleus, which contain an inner electron-dense mesh. (D) An S1 cell showing a large accumulation of granules with a looser electron-dense mesh, in the apex. (E) An S2 cell with the cytoplasm almost filled with vacuoles containing a microgranular substance of moderate electron density. Scale bars represent: (B) 1 μm; (C) 500 nm; (D–E) 1 μm. Abbreviations: adh, adherent junction; bla, basal lamina; cil, cilia; dcg, dense-core granules; dtu, bundles of electron-dense tubules; gly, glycogen deposit; icc, intercellular canaliculi; ics, intercellular space; mug, mucinogen granules; mvb, multivesicular body; mvi, microvilli; neu, neurite bundle; nuc, cell nucleus; sep, septate junction; tep, thin epithelial projections; vac, vacuoles.

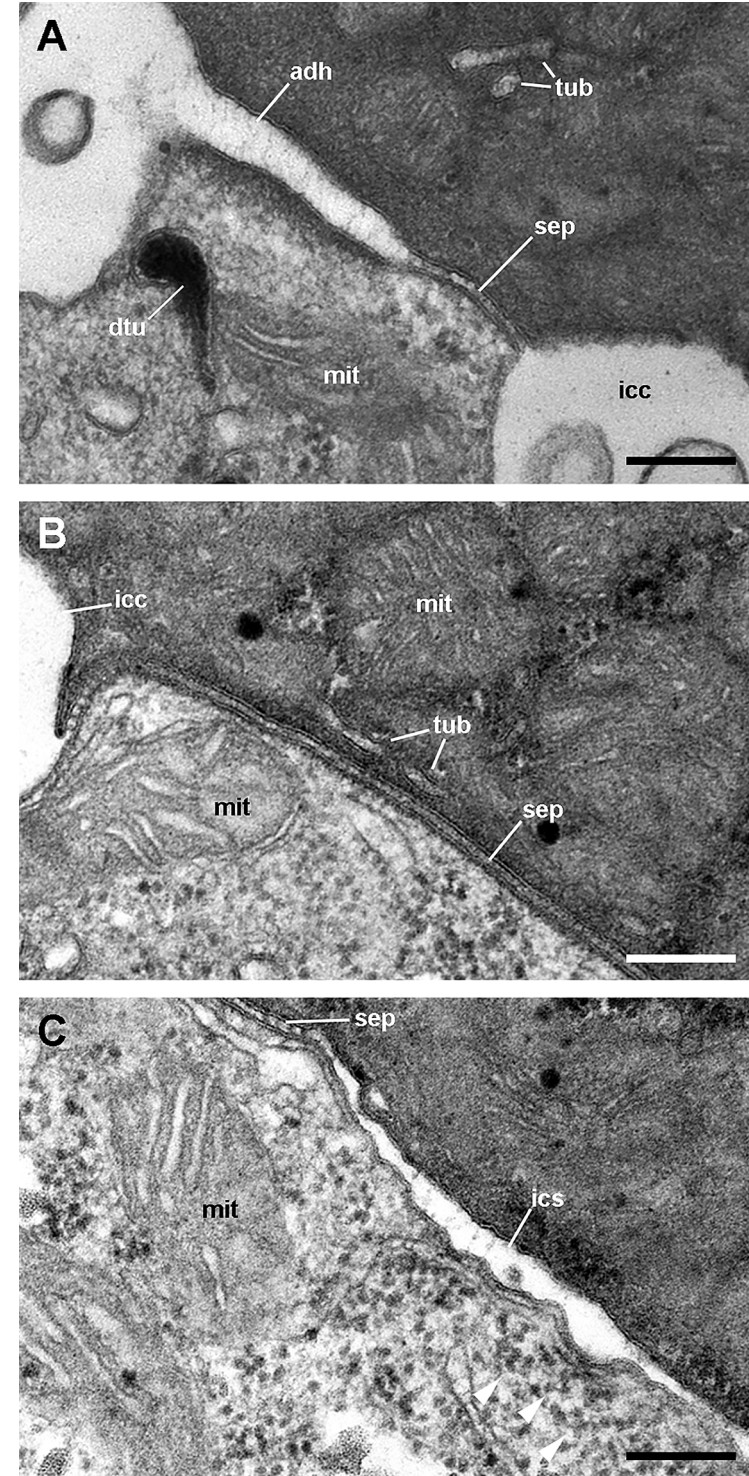

**Figure 12 Cell junctions in the gill epithelium (transmission electron microscopy).** (A) Apical adherent junction followed by a short septate junction and an intercellular canaliculum with some content. (B) A septate junction. (C) Widening of an intercellular space below the septate junction. Scale bars represent 200 nm. Abbreviations: adh, adherent junction; dtu, bundles of electron-dense tubule; icc, intercellular canaliculum; ics, intercellular space; mit, mitochondrion; sep, septate junction; tub, tubular system.

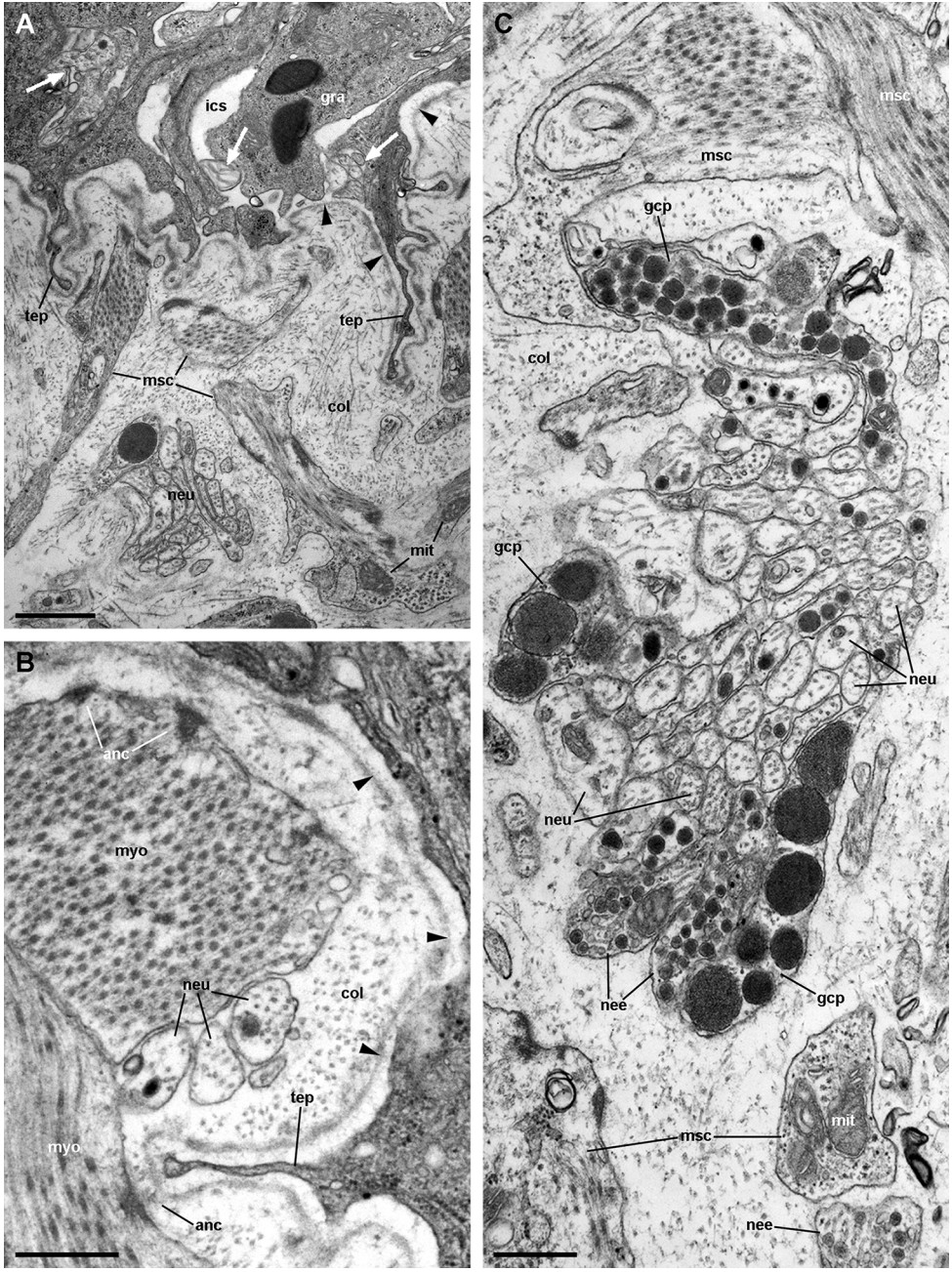

**Figure 13 Fibromuscular tissue and fine innervation of the gill leaflets (transmission electron microscopy).** (A) Overview of the basal domain of epithelial cells, together with a granulocyte in contact with the basal lamina (arrowheads). The underlying tissue exhibits a neurite/glial bundle and trabecular fibromuscular cells with inner myofibrils and electron-dense anchoring junctions with the collagen matrix and the basal lamina. Also notice some intraepithelial neurites (white arrows). (B) Detail of anchoring junctions showing the external brush-like plaque and the internal amorphous electron-dense layer. (C) Detail of a neurite bundle showing neurites associated with glial cells' processes containing granules of different sizes and electron density. Glial cells, or rarely uncovered neurite bundles, are in contact with muscle fibres or trabeculae. Scale bars represent: (A) 1 µm; (B) 500 nm; (C) 500 nm. Abbreviations: anc; anchoring junction; col, collagen matrix; gcp, glial cell process; gra, granulocyte; ics, intercellular space; mit, mitochondria; msc, muscle cells; myo, myofibrils; nee, presumptive nerve endings; neu, neurite bundles; tep, thin epithelial projection.

Thus, in the gill leaflets of *Pomacea canaliculata*, deoxygenated blood coming from the *afferent branchial vessel* would reach the *basal* and *marginal leaflet sinuses* and would distribute through the *laminar leaflet sinus*, where incomplete oxygenation should happen (Fig. 2F). In this way, partially oxygenated blood would converge either to the *efferent pulmobranchial vessel* or the *ventral afferent pulmonary vessel*, as shown in Fig. 2F. The fact that partially oxygenated blood went directly to the lung floor would prove to be useful to complete the $O_2$ uptake when the animal is submerged, because the gill might be insufficient to do that. Moreover, when on land, the collapse of leaflets and their laminar sinuses would force blood to follow the *basal* or *marginal sinuses* converging in the *ventral afferent pulmonary vessel* (Fig. 2F). In this way, there would be a shunt to the lung circulation, where oxygenation should reach its maximum (*Maina, 1990*).

Alternatively, the gill may function as a ventilator, generating the left-to-right current that exposes water to the wall of the pallial cavity, where some gas exchange might occur, as suggested by *Haszprunar (1992)*. Nonetheless, both the branchial and pallial surfaces together are still insufficient in *Pomacea canaliculata* for respiration under water, because this species still needs air to ventilate the lung for adequate oxygen supply (*Seuffert & Martín, 2010*). Indeed, *Seuffert & Martín (2010)* showed that *Pomacea canaliculata*'s microdistribution in the field mostly restricted to 2–4 m from the nearest emergent substratum, and that hindering aerial respiration negatively affected its survivorship in aquaria.

## The gill as an ionic/osmotic regulator

*Pomacea canaliculata* is a hyperosmotic and hyperionic regulator and, like in many other freshwater animals, its gill may be involved in this regulation. Indeed, the gill of *Pomacea canaliculata* has a high ion-ATPase activity, which suggests it is a site of ion uptake from the surrounding water, while its ureter would be a site of ion reabsorption from the primary urine (*Taylor & Andrews, 1987*).

As discussed above, the gill epithelium of *Pomacea canaliculata* (Fig. 4) is characterised by tall columnar cells (>20 μm) with apical specialisations, numerous mitochondria, and basolateral infoldings of the plasma membrane that enclose broad and presumably dynamic intercellular spaces. Additionally, these highly polarised cells have well-developed endomembrane systems and glycogen deposits (see Figs. 6 and 7). In contrast, the gill epithelium of *M. cornuarietis* is lower (~10 μm) than that of *Pomacea canaliculata* and has no intercellular spaces (*Lutfy & Demian, 1965*). It should be kept in mind that this species is more dependent on water than *Pomacea canaliculata* (e.g. it lays gelatinous eggs underwater; *Hayes et al., 2009a*), and thus relies mainly on its gill for respiration.

There are two main morphological types of presumptive ion-transporting cells in *Pomacea canaliculata*, which may be equivalent to the α and β mitochondria-rich cells found in freshwater teleost fishes (for a review, see *Evans, Piermarini & Choe, 2005*; *Wilson & Laurent, 2002*) and to the 'fibrillar' and 'tubulovesicular' types found in amphibians (*Lewinson, Rosenberg & Warburg, 1987*). Both α- and β-cells of *Pomacea canaliculata* are indeed mitochondria-rich cells. Like those of fishes, α-cells have an electron-lucent cytoplasm, an apical membrane slightly concave with few and short apical specialisations, and a well-developed subapical vesicular system (Figs. 6 and 7A).

This cell type also resembles the amphibian 'fibrillar cells' because of its membrane-bound bundles of electron-dense filaments/tubules (Fig. 7B). On the other hand, β-cells have an electron-dense cytoplasm and complex apical specialisations of the plasma membrane (Figs. 6 and 7D), as fish β-cells do. They also have a well-developed tubulovesicular system that almost fills the cytoplasm between mitochondria (Fig. 7E), in close resemblance to the 'tubulovesicular' cell type of amphibians (*Lewinson, Rosenberg & Warburg, 1987*). It should be noted that the highlighted similarities between fish, amphibian and *Pomacea canaliculata*'s mitochondria-rich cell types suggest some similar regulatory mechanisms.

Thus, the features found in the gill epithelium of *Pomacea canaliculata* contrast with those of the gill respiratory epithelia found in some other molluscan (*Fischer et al., 1990*; *Gregory, George & McClurg, 1996*; *Le Pennec, Beninger & Herry, 1988*; *Manganaro et al., 2012*; *Nuwayhid, Davies & Elder, 1978*) and non-molluscan taxa (*Evans, Piermarini & Choe, 2005*; *Luquet et al., 2002*) that are more dependent on water breathing, and which show cubic or squamous cells, with low nuclear/cytoplasmic ratios and a low content of mitochondria and other organelles. In turn, the gill structures found in *Pomacea canaliculata* are more similar to those of transporting epithelia (*Berridge & Oschman, 2012*), such as those of the vertebrate small intestine (*Flik & Verbost, 1993*), gallbladder (*Housset et al., 2016*), and renal tubules (*Yu & Chir, 2017*), and of the ionic/osmotic regulatory epithelia in the gills of crustaceans (*Luquet et al., 2002*; *McNamara & Faria, 2012*) and fishes (*McDonald, Cavdek & Ellis, 1991*). Taken together, these features may determine *Pomacea canaliculata* to be an obligate air-breathing species, whose gill structures seem better suited for ionic/osmotic regulation than for $O_2$ uptake.

## The gill as an immune barrier

In general, integumentary structures are the first barrier to microbial intruders, and as such, the gill is one of these structures preventing their access from the mantle cavity. In fact, *Pomacea canaliculata* must also cope with the diverse symbiotic organisms that frequently dwell in the mantle cavity (*Vega et al., 2006*). Mucus secretion and water currents may be unfavourable for the settling of many organisms (*Vega et al., 2006*). Abundant mucus is found covering the gill epithelium, which is likely secreted by the large number of secretory cells that occur in regions II and IV (Figs. 4 and 10). These cells are of two types, S1 and S2, that may correspond to the 'mucous' and 'gland' cell types in *M. cornuarietis*, respectively (*Lutfy & Demian, 1965*).

However, the gill is also a potential barrier because of its position in the circulation, as are the kidney and lung in *Pomacea canaliculata* (*Rodriguez et al., 2018*). Indeed, most blood coming from the cephalopodal mass and the visceral hump has to pass through the gill before reaching the heart to re-enter the general circulation. Thus, the gill itself, beyond its role as part of the integumentary barrier, may also function as a filter for blood-borne microorganisms.

The conspicuous occurrence of granulocytes amongst epithelial intercellular spaces (Figs. 4 and 8) suggests these cells would serve in immune defense in this organ. Granulocytes are also frequent in the gill of *M. cornuarietis*, but they appear subepithelial in the drawings of *Lutfy & Demian (1965)* rather than intraepithelial. The occurrence

of immunocompetent cells within epithelial intercellular spaces is a widespread feature amongst the gills of bivalves (*De Oliveira David, Salaroli & Fontanetti, 2008*; *Gregory, George & McClurg, 1996*) and fishes (*Hughes & Morgan, 1973*). The granulocytes found in the gill were larger than those found in the general circulation of *Pomacea canaliculata* and, in spite of being the less frequent cell type in the circulation (see *Cueto et al., 2015*), the intercellular spaces had only granulocytes within them.

There is evidence of a kind of 'compound exocytosis' (*Pickett & Edwardson, 2006*) leading to granulocyte degranulation in *Pomacea canaliculata* (*Cueto et al., 2015*). Granulocyte degranulation in bivalves (*Ciacci et al., 2009*; *Cheng et al., 1975*; *Foley & Cheng, 1977*; *Mohandas, Cheng & Cheng, 1985*; *Rebelo et al., 2013*) has been related to the release of lysozyme and other hydrolytic enzymes that may kill bacteria and fungi, and *Ottaviani (1991)* has reported lysozyme from haemocytes of a gastropod. Therefore, it is likely that granulocytes occurring in the intercellular spaces of the gill epithelium are there serving a defensive role in *Pomacea canaliculata*.

## Nervous control of the gill

The gill is mainly innervated from the *supraoesophageal ganglion* and the *accessory visceral ganglion* found in *Pomacea canaliculata* (see Fig. 3A), which are part of the ganglionar visceral loop (*Chase, 2002*) that also includes the suboesophageal and the visceral ganglia (*Berthold, 1991*; *Hylton Scott, 1957*). The origin of the gill innervation in the *supraoesophageal* and *accessory visceral ganglia* supports the view of the adult's gill as the post-torsional left gill (*Lindberg & Sigwart, 2015*), but which has been displaced to the right by the development of the lung (*Koch, Winik & Castro-Vazquez, 2009*). The *branchial nerve* also innervates the osphradium and the muscular region of the lung that surrounds the pneumostome (C Rodriguez, GI Prieto, IA Vega & A Castro-Vazquez, 2019, unpublished data). The osphradium has been shown to sense ionic or $O_2/CO_2$ levels in water, amongst other chemosensory functions in several gastropods (*Lindberg & Sigwart, 2015*). It also may be involved in the switch between the behavioural modes of branchial and lung respiration (C Rodriguez, GI Prieto, IA Vega & A Castro-Vazquez, 2019, unpublished data), and in the regulation of the gill ionic/osmotic functions in *Pomacea canaliculata*.

We have not found any neuroepithelial cells similar to those found in fish gills (*Bailly, Dunel-Erb & Laurent, 1992*; *Dunel-Erb, Bailly & Laurent, 1982*; *Jonz & Nurse, 2003*). The neurite supply to the gill leaflets is rich and spread in the connective tissue, the laminar sinus cells, and the epithelial basolateral domain (see Fig. 3), as has been described in the gill filaments of *Anodonta* (*Nakao, 1975*) and in the gill leaflets of many fishes (*Jonz & Zaccone, 2009*). It also includes glial cell processes (Fig. 12C) similar to those referred by *Nicaise (1973)* in heterobranchs.

The sensory information coming from the osphradium and/or the gill epithelium may be integrated in the visceral loop and may trigger different responses through efferent pathways. For example, controlling the state of the intercellular spaces could either increase or decrease the epithelium permeability, thus regulating respiratory or ionic/osmotic functions, as there has been described in fish gills (*Jonz & Nurse, 2006*).

Efferent pathways may also be involved in the regulation of vascular resistance through the gill leaflets by altering the stretching of trabecular cells, as has been proposed for bivalves (*Nakao, 1975*) and as it occurs in pillar cells of fish gills (*Jonz & Zaccone, 2009*). Indeed, numerous neurites were often found in close contact with these modified muscle cells (see Fig. 12). Finally, other motor innervation would involve that associated with the muscular bundle along the efferent border of the gill leaflet (see Fig. 6).

## The evolution of amphibiousness: the family Ampullariidae as a case study

The family Ampullariidae has been proposed as a model for evolutionary biology because of its long evolutionary history that traces back to the Jurassic (~160 million years ago), wide geographic distribution (through Africa, Asia and the Americas), and high diversity (~120 currently valid species in nine genera) (*Hayes et al., 2009a*). These characteristics, along with the different degrees of air dependence ampullariids show, make this family an interesting model to study the evolution of amphibiousness (*Hayes et al., 2015*). Important advances have been made in elucidating the evolution of traits related to amphibiousness in Ampullariidae. In particular, the evolution of aerial oviposition has received considerable attention (*Hayes et al., 2009a*; *Ip et al., 2019*; *Mu et al., 2017*; *Sun et al., 2019*). However, a comparative study on the morphology, function and development of the respiratory organs amongst the Ampullariidae is still lacking.

As mentioned above, the development of a lung has allowed a shift in the biological role of the gill to the detriment of its capacity for oxygen uptake in bimodal-breathing crustaceans and fishes. Our results suggest this may also be the case amongst the Ampullariidae and encourage the search for similar patterns through the comparison of the respiratory organs, and their relative functions, between ampullariid species with different degrees of air dependence.

Finally, it is worth emphasising the convergence of gill structures of *Pomacea canaliculata* with those in phylogenetically distant taxa, such as arthropods (*Farrelly & Greenaway, 1987*; *Luquet et al., 2002*; *Maina, 1990*) and fishes (*Evans, Piermarini & Choe, 2005*; *Laurent & Dunel, 1980*; *Low, Lane & Ip, 1988*; *McDonald, Cavdek & Ellis, 1991*). Indeed, these may be cases of phenotypic convergence in which similar genetic mechanisms, such as the existence of conserved master regulators, can lead to convergence in form and function in independent and often distant lineages (*Stern, 2013*). The existence of master regulators, such as the Hox and ParaHox genes, has been shown in ampullariids (*Sun et al., 2019*) and in representatives of other classes of molluscs (*De Oliveira et al., 2016*). Future studies may be aimed at elucidating whether conserved master regulators are involved in the development of similar structures in the gill of *Pomacea canaliculata*.

## CONCLUSIONS

1. We have confirmed interpretations of preceding authors regarding the vasculature and innervation of the gill of *Pomacea canaliculata* and their implications. Namely, (a) that the gill vasculature is connected in series with that of the lung, in such a way that

blood may complete oxygenation in the latter organ, and (b) that the origin of the gill innervation in the main and *accessory supraoesophageal ganglia* supports the view of the adult's gill as the post-torsional left gill, but which has been displaced to the right by the development of the lung.

2. When the animal is under water, the gill surface potentially available for gas exchange is large but is covered by a rather thick epithelium (>20 μm) with no cubic or squamous cells as in the gills of crustaceans and fishes. Ultrastructural evidence suggests that the only structures that perhaps facilitate oxygen uptake would be those involved in the paracellular pathway.

3. Also, in case the branchial leaflets collapse when the animal is out of the water, blood may bypass the leaflets and go directly to the lung through a shunt formed between the *afferent branchial vessel* and the *ventral afferent pulmonary vessel* through the *basal branchial sinuses*.

4. The leaflet architecture is uniform throughout the whole gill, that is, there is not a respiratory and ionic regionalisation of the gill as there occurs in other taxa (e.g. crustaceans). However, four regions may be recognised within each leaflet, each presumably associated with specific functions: region I with ionic/osmotic regulation, regions II and IV with mucous secretion, regions III and IV with water circulation, and region III to gas exchange through the paracellular pathway.

5. Our findings showed that the gill epithelium has features of a transporting epithelium rather than a respiratory one. Specifically, the branchial epithelium has (1) developed apical specialisations and basolateral infoldings, (2) occluding junctions, (3) extensive, and likely dynamic, intercellular spaces, (4) a high density of mitochondria, and (5) an underlying rich nerve supply. Altogether, these features suggest the gill in *Pomacea canaliculata* would be more suitable for ionic/osmotic regulation than for oxygen uptake, which partly explains why *Pomacea canaliculata* is an obligate air-breathing species.

6. The gill may function as an immune barrier by secreting mucus to prevent the access of intruders from the mantle cavity, but also to prevent the spread of blood-borne microorganisms, in which granulocytes may participate.

## ACKNOWLEDGEMENTS

The authors appreciate the generous help of Elisa Bocanegra, Sergio A. Carminati, Norberto F. Domizio, Mabel Fóscolo, María Silvina Lassa, and María Paula López.

### Funding

This work was funded by Fondo para la Investigación Científica y Tecnológica (FONCYT), grant PICT 2013-1190 (to Alfredo Castro-Vazquez) and by Universidad Nacional de Cuyo, grants RCS05712015 (to Israel Aníbal Vega) and 06/M097

(to Alfredo Castro-Vazquez). The funders had no role in study design, data collection and analysis, decision to publish, or preparation of the manuscript.

### Grant Disclosures
The following grant information was disclosed by the authors:
Fondo para la Investigación Científica y Tecnológica (FONCYT): PICT 2013-1190.
Universidad Nacional de Cuyo: RCS05712015 and 06/M097.

### Competing Interests
The authors declare that they have no competing interests.

### Author Contributions
- Cristian Rodriguez conceived and designed the experiments, performed the experiments, analysed the data, prepared figures and/or tables, authored or reviewed drafts of the paper, approved the final draft.
- Guido I. Prieto conceived and designed the experiments, performed the experiments, analysed the data, prepared figures and/or tables, authored or reviewed drafts of the paper, approved the final draft.
- Israel A. Vega conceived and designed the experiments, analysed the data, contributed reagents/materials/analysis tools, authored or reviewed drafts of the paper, approved the final draft.
- Alfredo Castro-Vazquez conceived and designed the experiments, analysed the data, contributed reagents/materials/analysis tools, prepared figures and/or tables, authored or reviewed drafts of the paper, approved the final draft.

### Ethics
The following information was supplied relating to ethical approvals (i.e. approving body and any reference numbers):

The Institutional Committee for the Care and Use of Laboratory Animals (Comité Institucional para el Cuidado y Uso de Animales de Laboratorio (CICUAL), Facultad de Ciencias Médicas, Universidad Nacional de Cuyo) approved procedures for snail culture, sacrifice, and tissue sampling (Approval Protocol No 55/2015).

### Data Availability
Raw data is available at Figshare: Rodriguez, Cristian (2019): Gill structures of *Pomacea canaliculata*. figshare. Figure. DOI m9.figshare.8143790.v1.

A 3D model of the blood system of the gill is available at MorphoSource, specimen identifier: S19912.

https://www.morphosource.org/Detail/SpecimenDetail/Show/specimen_id/19912.

### Supplemental Information
Supplemental information for this article can be found online at http://dx.doi.org/10.7717/peerj.7342#supplemental-information.

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
