# Peer review of "Functional and evolutionary perspectives on gill structures of an obligate air-breathing, aquatic snail"

_PeerJ, doi:10.7717/peerj.7342_

## Round 0.1 · original submission · Major Revisions

As you will see, your interesting manuscript has been read and commented upon by two expert reviewers. They find the study exceptional or at least potentially important, have various suggestions for additions and improvements, and come to rather different recommendations concerning its publication.

I believe both of them raise valid and important points and I am returning your manuscript for revision. In response to Reviewer 1, please have an especially close look at the terminology you employ for organ structures. I agree with Reviewer 2 in that the discussion would benefit from covering more of the suggested literature. In your ‘rebuttal’ letter, please describe how you dealt with each of the items in the two reviews.

Reviewer 1 ·

Basic reporting

This is an exceptional study on the anatomy and fine structure of the gill of the invasive apple snail, Pomacea canaliculata. The authors bring together 3D reconstruction with histology, SEM and TEM to bring valuable new insight into the structure and function of the gill in this family, and the implications for the evolution of amphibiousness. The manuscript is clearly and carefully written, with professional English used throughout. The anatomical descriptions are straightforward and easy to understand. They have taken great care to ensure that the language is clear and precise. The literature cited is comprehensive. The figures are of outstanding quality and clarity. Goals are clearly defined, and the authors have thoughtfully placed their study into context and discussed the implications of their findings.

My main criticism is in their occasional introduction of terminology that differs from established terms in caenogastropod anatomy, and especially compared to those used by Andrews in her exemplary study on the functional anatomy of the mantle cavity, kidney and blood system of ampullariids. This creates confusion, makes it difficult to compare the results of different studies, and obscures homology with other caenogastropods (e.g., ureter, efferent ureteral vessel). In some cases, the terms they have introduced are not adequately descriptive of the feature they describe (e.g., mantle fold; see below). I would also like to see a discussion of any differences in interpretation regarding circulation in the mantle roof between this study and that of Andrews (1965), highlighting new results and insights of this study.

Experimental design

no comment.

Validity of the findings

see above.

Additional comments

Comments and suggestions:
In several instances, the authors use colloquial language to describe the shape of structures (e.g. comparisons to a shark’s fin or to cauliflower) which I would remove.

Lines 73-74: there have been TEM studies of vent caenogastropod gills (ie Provannidae) that have symbionts.

Lines 76-77: “because we were stricken by the morphological parallelisms found between the gill of P. canaliculata and those of somewhat phylogenetically distant taxa.”: Delete

Line 91: dilute Bouin’s fluid

Line 108: “material was fixed and embedded as described above”; two methods were described above. Presumably you are referring to embedding in histoplast here.

Line 119: dilute Bouin’s fluid

Line 135: Referring to the anterior chamber of the kidney as the ureter is not standard terminology in the anatomical literature of ampullariids, and confuses the homology of this structure. For example, the ureter of viviparids is a de novo extension of the kidney into the mantle roof alongside the intestine. What you have termed ureter, is an anterior kidney chamber. Many caenogastropods are known to have differentiated regions of the true kidney that are specialized for different functions. The term ureter has been used uncritically in studies of caenogastropods, and I would advise against perpetuating this practice.

Line 143: among “siphonate” gastropods, there are those with a true siphon (=siphonate clade), and those with simply an extended mantle edge (Ponder et al 2008). In both cases, in contrast to ampullariids, the siphon is derived from the mantle roof.

Line 144: why is pallial fold in quotation marks? This is certainly more accurate than “mantle fold”, as it is termed in Figure 1, because it is not a fold in the mantle, it is a fold on the floor of the mantle cavity. I also don’t understand the need to introduce different terminology, when the existing terminology is perfectly adequate.

Lines 148-149: rather than groove, I would suggest channel or something similar. “Groove” suggests to me a much more narrow structural definition (ie. an indentation or furrow).

Line 156: The 3D rendering shows (verb agreement)

Line 157: efferent ureteral vessel. Again, this terminology relating to the “ureter” is obscuring homology of these structures with other caenogastropods. This term has never been used in descriptions of gastropod vasculature, and obscures comparisons with homologous structures in other caenogastropods.

Line 174: supraintestinal. More typically referred to as supraesophageal in the recent literature.

Line 175 accessory supraintestinal ganglion. Given its position, it seems more appropriate to term this structure an accessory visceral ganglion. The presence of accessory ganglia on the visceral loop in caenogastropods is well known.

Line 176: viscero-supraintestinal connective. This is technically incorrect, as the connective joins the supraintestinal and “accessory supraintestinal” ganglia. This is the supraesophageal portion of the visceral loop.

Line 194: what kind of shark? Preferable to use standard geometric terms to describe the shape.

Lines 222-223: “which somehow remind cauliflowers under the scanning electron microscope”: Delete.

Line 242: in region I

Line 266: are also found in the

Line 275: these discontinuities connect

Lines 307-309: “Indeed, in the extreme case of an aquatic animal being brought to air, the gill leaflets tend to stick together thus decreasing the surface area for the gas exchange, and hence, leading the animal to asphyxia (Maina 2002b).”. But the presence of rods in other gastropods means that their presence is not related to facilitating gas exchange while emergent. How do you explain that?

Line 328: The classification of Bouchet et al (2005) has been superseded. In the most recent version (2017), the Architaenioglossa is a grade.

Line 393: “P .canaliculata”. The space needs to come after the period.

Line 418: dwell in the

Line 440: granulocytes occurring in the

Line 470: “The bases of terrestriality”: it should be “basis” not “bases”, and I would argue they aren’t terrestrial. Replace with “The evolution of amphibiousness” or something similar.

Lines 478-479: has received considerable attention

Lines 503-504: “supports the view of the adult’s gill as the post-torsional left gill”: this is the first mention of this in the manuscript. It shouldn’t appear for the first time in the conclusions, but should be mentioned elsewhere in the main body of the results or discussion.

Figure 1: tes, should be testes, not testicle. Upo, should be nephropore, the standard term for the opening of the kidney to the mantle cavity. Pallial fold, rather than “mantle” fold.

Figure 2: What is “the dissection piece”? “Limiting” what with the lung? The meaning here is obscure. Epb is not in the list of abbreviations. “Rectal sinuses” is not correct. There are not multiple sinuses. There is a single rectal sinus and channels that drain it. 2B. I would rotate this figure 180, so that anterior is below, as the other images in this figure.

Figure 3:
Major ganglia vs central ganglia. “Central” implies something positional.
Penial vs penile.
Vsc. As mentioned above, technically this is the accessory supraintestinal-supraintestinal connective. Typically, and more simply, referred to as the supraesophageal part of the visceral loop.

Figure 4:
Diagram of the four regions of the gill leaflet. Presumably these observations are generally applicable to all gill leaflets, not just this one. Change throughout (see e.g. legends for Figs 6, 7, 9, 10)

Figure 5:
“Piece of dissection”: there has to be a more elegant way of saying this. Apparently similar usage to that in the legend for Figure 2 which I found confusing.
α, alpha-cell; β, beta-cell. On subsequent figures, the labels for these cells are α-c and β-c. This should be standardized.

Figure 6:
ics is not in the list of abbreviations.

Figure 7:
Myeloid bodies, and fibrogranular material, not “figures”

Figure 8:
Generally, the black labels are very difficult to make out on the dark background; suggest to use white.
8C. …connect the intercellular spaces….

Figure 10:
It seems far more common to use “vacuoles” rather than “vacuolae”, which I had never seen used before.
Again, some of the black labels are very difficult to make out on the dark background; suggest to use white.

Fig. 11:
11C: “mit” is very difficult to see.

Fig. 12:
“cav, caveolae”: this term appears nowhere else in the manuscript.

Fig. 13:
Myeloid bodies, not figures.

Table 1: Cell types and other features of the gill leaflet regions

·

Basic reporting

This is an ambitious study on the (fine-) structure of the gill of an apple snail. It has been carefully carried out with an array of sophisticated methods including LM sectioning plus computerized 3D treatment, SEM and TEM. The results bear the potential for an important publication.
However, at the present stage the MS cannot be recommended for publication because of a number of shortcomings.

Experimental design

This morphological investigation does not contain experiments. The examinations have been carried out carefully with sophisticated methology.

Validity of the findings

The findings are a valuable contribution to the knowledge of gill structure of ampullariids and the extremely large group of caenogastrops in general, which surprisingly have never been investigated in this respect with fine structural techniques before.

Additional comments

I list the most important flaws in the following in form of recommendations for improvement for a future submission:

* Improve the structure descriptions in the results:
The images are of really sound quality but they should be referenced much more intensely in the results text. For example, when a skeletal rod is first mentioned (lines 180-181) there is referred only to Fig. 3, where it is hardly discernible, but the structure is much better visible in Fig. 13.

* Disentangle Results and discussion:
For example, do not give comparisons with literature references in the results, like in lines 260/261. Vice versa, the skeletal rod description in the discussion (lines 328-330) provides details not anywhere found in the results. So, these are details that must be placed in the results.

* Get back to literature
This is the most important problem of the MS; quite some important papers were overlooked/ignored.

These studies
Demian, E. S. (1965). The respiratory system and the mechanisms of respiration in Marisa cornuarietis (L.). Arkiv Für Zoologi, 17(8), 539–560.
Lutfy, R. G., & Demian, E. S. (1965). The histology of the respiratory organs of Marisa cornuarietis (L.). Arkiv För Zoologi, 18(5), 51–71.
on a closely related species must be consulted and quoted as (basis and) previous treatment of the topic.

For the intro (function, organization, homology) on Gastropod “lungs” these papers should be consulted:
Ruthensteiner, B. (1997). Homology of the pallial and pulmonary cavity of gastropods. In Journal of Molluscan Studies (Vol. 63, pp. 353–367).
Lindberg, D. R., & Ponder, W. F. (2001). The influence of classification on the evolutionary interpretation of structure - A re-evaluation of the evolution of the palliai cavity of gastropod molluscs. Organisms Diversity and Evolution, 1(4), 273–299.
These would, for example, show that there are (1) more cases of „lungs“ in gastropods (e.g. all terrestrial prosobranchs) and that these structures are not necessarily homologous (e.g. pumonate lung vs. ampullariid lung).

Also, the original literature source for the finding of “skeletal rods” in ampullariids should be searched for. In the MS there is only secondary literature quoted (e.g. Haszprunar, 1988 [source missing in there too]). This is important for the interpretation/(yet missing) discussion on the “skeletal rods”, which in fact – as clearly evident from the findings in the MS – represent muscles, which ARE NOT skeletal elements (see also Lutfy & Demian, 1965).

Here a hint on the discussion on the gill function: The authors acknowledge the idea (Andrews, 1965) of active water circulation be the gill ciliation but are puzzled by the absence of a respiratory epithelium in the gills. Why not consider a main function of water circulation by the gill and oxygen uptake elsewhere (wall of pallial cavity) as suggested by e.g. Haszprunar 1991 for many molluscs.
Haszprunar, G. (1992). The first molluscs - small animals. Bolletino Di Zoologia, 59(1), 1–16.

In total, the discussion needs to be largely rewritten. In general, I recommend undertaking comparisons with other molluscs (e.g. tons of studies on bivalves) rather than with distantly related taxa (crustaceans, vertebrates) to remain on a solid basis.

Some more remarks can be found in the annotated MS.

With best regards,

Bernhard Ruthensteiner

---

## Round 0.2 · Minor Revisions

As you will see, both of the reviewers who had commented on the earlier manuscript version suggest a few minor changes/clarifications. I especially agree with Reviewer 2 that the manuscript will benefit from a clearer distinction between supporting rods in other taxa and the situation found in this species. I look forward to seeing the revised version.

Reviewer 1 ·

Basic reporting

No comment.

Experimental design

No comment.

Validity of the findings

No comment.

Additional comments

I am very satisfied with the revisions to the manuscript. I find the discussion, in particular, to be very well done and much improved. The conclusions are clearly stated and concise, and succinctly summarize the main findings of this study while placing them in a broader context.
I have a few minor editorial suggestions for the authors to take into consideration:
Line 60: extends into the roof of the pallial cavity
Line 134: “Then, the profile of identified structures (objects) were drawn with the mouse using the Trace tool. These traces were used for the 3D visualisation of object”: This process is typically called “segmentation”, in this case manual segmentation. Is the terminology used herein software specific?
Line 148: “in which the main afferent and efferent vessels run”: reference unclear, could be “bases” or “roof”.
Line 150: “Based on their relative position with respect to the blood flow”: This isn’t untrue, but wouldn’t it be more practical (and more conventional) to define the afferent and efferent borders relative to their orientation in the mantle cavity, and hence to water flow?
Line 152: and therefore to the inner mantle? I would take the “outer mantle” to refer to the external body wall of the mantle roof.
Line 154: close to the base of the siphon.
Line 187: presumably the lung roof is the external wall of the mantle?
Line 191: located along the supraoesophageal portion
Line 194: has not been described for P. canaliculata
Line 309: have no abfrontal
Line 313: requires both a large and thin
Line 315: to permit adequate
Line 340: referred to the existence
Line 347: in molluscan gills
Line 373: facilitating O2 extraction from the water
Line 421: “(e.g., it lays gelatinous eggs underwater” – this is repeated twice.
Line 467: immune surveillance: this is a very sophisticated notion for the immune response of gastropods, and one that is distinctly anthropormorphic given the widespread use of this term in human studies.
Line 481: delete “against intruders”
Fig. 1B: “below the outer mantle”: what does this mean?
Supplementary Table 1: I would suggest to restrict the comparisons to those where the interpretation is different, or where confusion may occur due to the use of different terminology. Examples where only the abbreviations differ are not useful.

·

Basic reporting

2nd Review of the Manuscript
Functional and evolutionary perspectives on gill structures of an obligate air-breathing, aquatic snail
by C.Rodriguez, G.I. Prieto, I.A. Vega Corresp and A. Castro-Vazquez

The MS is improved in many ways.
I can recommend publication if the issues annotated in the PDF MS are considered/treated. I must insist in modifying the terminology (“rod”) and discussion of/on the gill muscle.

With best regards,

Bernhard Ruthensteiner

Experimental design

Is o.k.

Validity of the findings

Is o.k.

---

## Round 0.3 · accepted · Accept

Thank you for responding to the reviews so thoroughly. I am happy to accept it in its current form and look forward to the publication of this very fine article.